# Event-related modulation of alpha rhythm explains the auditory P300-evoked response in EEG

**Alina Studenova[1,2]\*, Carina Forster[1,3], Denis Alexander Engemann[1,4], Tilman Hensch[5,6,7], Christian Sanders[5,7], Nicole Mauche[7], Ulrich Hegerl[8], Markus Loffler[5,9], Arno Villringer[1,10], Vadim Nikulin[1,11]**

[1]Department of Neurology, Max Planck Institute for Human Cognitive and Brain Sciences, Leipzig, Germany; [2]Max Planck School of Cognition, Leipzig, Germany; [3]Bernstein Center for Computational Neuroscience, Charité – Universitätsmedizin Berlin, Berlin, Germany; [4]Roche Pharma Research and Early Development, Neuroscience and Rare Diseases, Roche Innovation Center Basel, F. Hoffmann–La Roche Ltd., Basel, Switzerland; [5]LIFE – Leipzig Research Center for Civilization Diseases, University of Leipzig, Leipzig, Germany; [6]Department of Psychology, IU International University of Applied Sciences, Erfurt, Germany; [7]Department of Psychiatry and Psychotherapy, University of Leipzig Medical Center, Leipzig, Germany; [8]Department of Psychiatry, Psychosomatics and Psychotherapy, Goethe University Frankfurt, Frankfurt, Germany; [9]Institute for Medical Informatics, Statistics and Epidemiology (IMISE), University of Leipzig, Leipzig, Germany; [10]Clinic for Cognitive Neurology, University Hospital Leipzig, Leipzig, Germany; [11]Bernstein Center for Computational Neuroscience Berlin, Berlin, Germany

*For correspondence: studenova@cbs.mpg.de

**Abstract** Evoked responses and oscillations represent two major electrophysiological phenomena in the human brain yet the link between them remains rather obscure. Here we show how most frequently studied EEG signals: the P300-evoked response and alpha oscillations (8–12 Hz) can be linked with the baseline-shift mechanism. This mechanism states that oscillations generate evoked responses if oscillations have a non-zero mean and their amplitude is modulated by the stimulus. Therefore, the following predictions should hold: (1) the temporal evolution of P300 and alpha amplitude is similar, (2) spatial localisations of the P300 and alpha amplitude modulation overlap, (3) oscillations are non-zero mean, (4) P300 and alpha amplitude correlate with cognitive scores in a similar fashion. To validate these predictions, we analysed the data set of elderly participants (N=2230, 60–82 years old), using (a) resting-state EEG recordings to quantify the mean of oscillations, (b) the event-related data, to extract parameters of P300 and alpha rhythm amplitude envelope. We showed that P300 is indeed linked to alpha rhythm, according to all four predictions. Our results provide an unifying view on the interdependency of evoked responses and neuronal oscillations and suggest that P300, at least partly, is generated by the modulation of alpha oscillations.

## eLife assessment

This is **valuable** study on the mechanistic relationship between two prominent events in post-stimulus EEG: alpha desynchronization and P300 that are known for their slow/relatively late build up. The sample size is substantial. The data are **compelling**, showing that the P300 can be explained by desynchronization of a non-zero mean alpha oscillations over posterior sites through

the baseline-shift model, at least partially. This makes a significant contribution to understanding and interpreting P300 generation (and possibly other ERP components) from concurrent changes in brain oscillations, with links to cognition.

## Introduction

P300 is one of the most extensively investigated evoked responses (ER) in electroencephalography (EEG) and magnetoencephalography (MEG). Over the years, P300 has been hypothesised to reflect a variety of functions, such as priming, cognitive processing, memory storage, context updating, resource allocation, etc. (*Polich and Kok, 1995*; *Polich, 2003*; *Verleger, 2020*), and there is an ongoing effort to understand its functions further through such constructs as information, expectancy, and capacity (*Verleger, 2020*). Usually, P300 is assessed with the oddball paradigm (auditory or visual), where participants have a task to detect a target (or rare, or deviant) stimulus in a train of standard (or frequent, or non-target) stimuli (*Luck, 2014*). Additionally, it is usual to speak about the P300 complex, involving the earlier frontal component P3a and the later parietal component P3b (*Linden, 2005*). Being aware of this forking terminology, in the following, we refer to P300 as the ER that occurs after the target stimulus and is different compared to the ER after the standard stimulus. Adding to the complexity of P300, the exact mechanism of P300 generation remains rather unknown (*Fell et al., 2004*; *Hanslmayr et al., 2007*; *Daly et al., 2009*; *Rawls et al., 2020*). In the present study, we investigate a possibility that P300 might be to some extent generated through a baseline-shift mechanism (BSM, *Nikulin et al., 2007*; *Mazaheri and Jensen, 2008*; *Iemi et al., 2019*; *Studenova et al., 2022*).

Apart from P300, the oddball target stimulus concurrently causes the attenuation of the alpha rhythm amplitude (8–12 Hz). The simultaneity of P300 and alpha rhythm modulation has been observed in numerous earlier and more recent studies, and in Appendix 1 we offer a short overview of these findings. We found 38 studies that presented results for a concomitant occurrence of P300 and alpha power (or amplitude). In 17 studies using EEG, results indicated an overlap in cortical regions of P300 and alpha amplitude decrease, as well as a similar time windows of their occurrence (*Peng et al., 2012*; *Chen et al., 2013*; *Dong et al., 2015*; *Shou and Ding, 2015*; *Tang et al., 2015*; *Wu et al., 2015*; *Fabi and Leuthold, 2017*; *López-Caneda et al., 2017*; *Vilà-Balló et al., 2017*; *Fabi and Leuthold, 2018*; *Michelini et al., 2018*; *Román-López et al., 2019*; *Kao et al., 2020*; *Yu et al., 2020*; *Zhang et al., 2020*; *Nikolin et al., 2021*; *Paolicelli et al., 2021*). Similar observations were made using MEG (*Ishii et al., 2009*). *Yordanova et al., 2001* and 14 more studies found similarities in location but not in the peak latencies (*Kolev et al., 2001*; *Kamarajan et al., 2006*; *Digiacomo et al., 2008*; *Krämer et al., 2011*; *Barutchu et al., 2013*; *Deiber et al., 2013*; *Kayser et al., 2014*; *Zarka et al., 2014*; *Deiber et al., 2015*; *Leroy et al., 2017*; *Liu et al., 2019*; *Martel et al., 2019*; *Faro et al., 2019*; *Espenhahn et al., 2020*). In a few studies, alpha modulation did not appear at all (*Kamarajan et al., 2004*; *Delval et al., 2018*) or the relationship between alpha oscillations and ER was not supported by cross-condition comparison (*Cooper et al., 2008*; *Lee et al., 2017*; *Tamura et al., 2016*). In general, we acknowledge that due to different ways of presenting results, sometimes it was difficult to tell whether the peak of P300 and the attenuation peak in the alpha amplitude correspond to each other. Nevertheless, the vast majority of studies confirmed the simultaneous occurrence of P300 and alpha amplitude decrease in several experimental paradigms, which in turn served as a basis for further investigation carried out in the present study.

The simultaneous presence of P300 and alpha amplitude modulation in the poststimulus window indicates that P300 can be partially generated through BSM (*Nikulin et al., 2007*; *Mazaheri and Jensen, 2008*; *Iemi et al., 2019*; *Studenova et al., 2022*). Previous research investigated whether the origin of P300 is due to an additive mechanism (*Fell et al., 2004*; *Wan et al., 2009*; *Herrmann et al., 2014*) or a phase-reset mechanism (*Fell et al., 2004*; *Daly et al., 2009*; *Wan et al., 2009*, but *Popp et al., 2019*), and the evidence for these mechanisms is far from converging. However, P300 has not yet been assessed with respect to BSM. In general theory, BSM links evoked activity and spontaneous oscillatory activity, stating that if oscillations are modulated by the stimulus presentation, this modulation will be mirrored in the low-frequency signal if oscillations have a non-zero mean (see *Figure 1*). In other words, the amplitude modulation of the oscillatory process affects the mean as well, which in turn leads to the deflection in the spectral range of modulation activity (with the frequency of modulation lying in a considerably lower range than the carrier frequency of oscillations

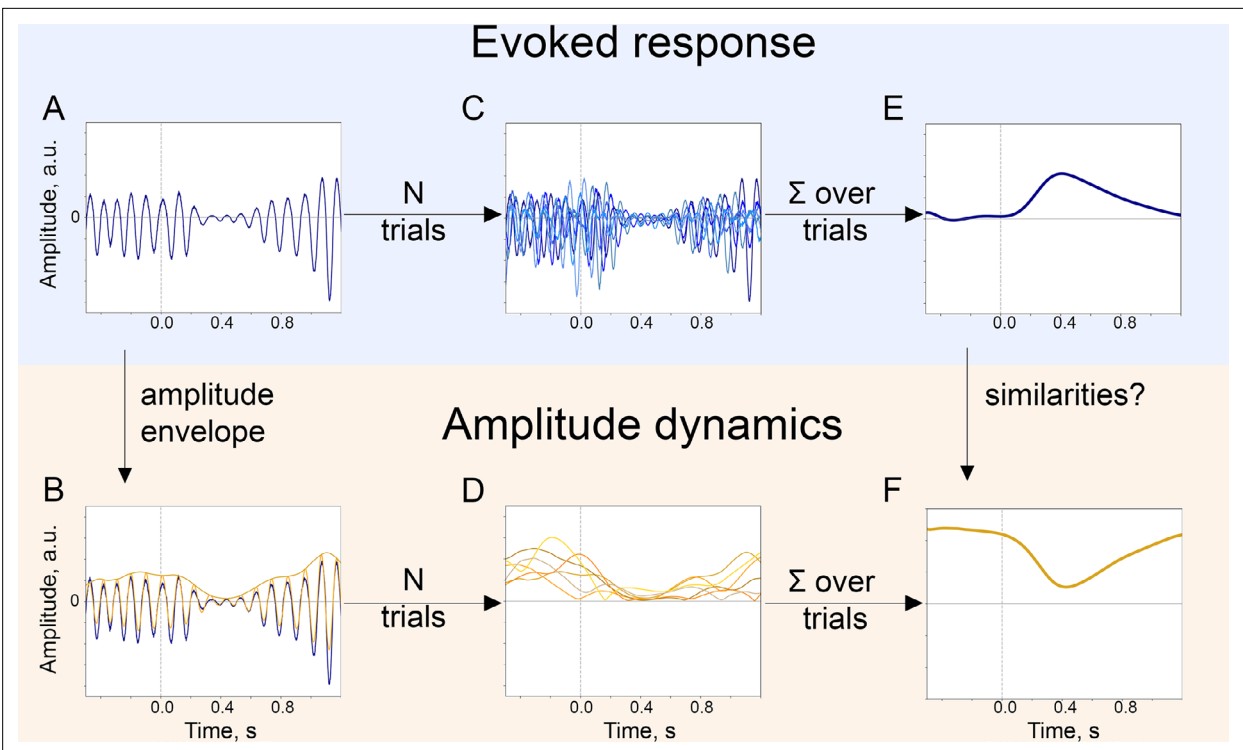

**Figure 1.** The baseline-shift mechanism (BSM) of evoked response (ER) generation. For a particular ER, probing the agreement with BSM would involve extracting both the ER and the oscillatory amplitude envelope. (**A**). The single-trial broadband signal. (**B**). The amplitude envelope of oscillations is extracted from a broadband signal of each trial. (**C**). To get a high signal-to-noise ER, usually multiple trials are acquired. Note that since oscillations have a negative mean, their attenuation would lead to the generation of an ER with a positive polarity (shown in E.). (**D**). Similarly, for each trial, the amplitude envelope is extracted. (**E**). Trials are averaged and, optionally, low-pass filtered to obtain an ER.( **F** ). Amplitude envelopes over trials are also averaged to obtain an estimate of the change in oscillatory amplitude in the poststimulus window. Here, we simulated the example of negative-mean oscillations giving rise to a positive-polarity ER.

themselves; for instance, if the oscillations' frequency range is 8–12 Hz, the modulation's frequency range is 0–3 Hz). In practice, when integrated over several periods, oscillations with a non-zero mean will show an average value different from zero that will scale with the amplitude of oscillations. Likewise, a non-zero mean implies that average values of the upper and lower half of the oscillatory cycle would be unequal. In *Figure 1*, negative-mean oscillations undergo a decrease in the amplitude in the poststimulus window and, according to BSM, the decrease in the amplitude of negative-mean oscillations creates an ER with a positive polarity. Here, oscillations are assumed to be ongoing, that is they are present before the stimulus onset. The polarity of the ER depends on the sign of the oscillatory mean and on the direction of modulation—an increase or decrease in the amplitude. The oscillatory mean of alpha oscillations has been shown to be present in biophysical model of alpha oscillations (*Studenova et al., 2022*) and several studies provided empirical evidence for the generation of ER through BSM in somatosensory (*Nikulin et al., 2007*) and visual (*Mazaheri and Jensen, 2008*; *Iemi et al., 2019*) domain. Since P300 coincides with the stimulus-triggered decrease in the alpha amplitude, it is reasonable to assess the compliance of P300 with BSM. Therefore, we hypothesised that P300 generation can at least partially be explained by the amplitude modulation of alpha oscillations, and in the following, we offer a systematic investigation of this hypothesis.

Assessing the compliance of ER to BSM requires the following four prerequisites: (1) demonstrating the similarity in the temporal evolution of both signals—P300 and alpha amplitude envelope—over time in the poststimulus interval, (2) showing the similarity of spatial locations of the neuronal processes giving rise to P300 and to alpha amplitude decrease, (3) linking the direction of ER with the direction of alpha amplitude modulation through the sign of oscillatory mean, (4) establishing similarity of a relation of ER/oscillations with external variables, such as cognitive performance. In the following sections, we present comprehensive evidence for the association of P300 with alpha oscillations using

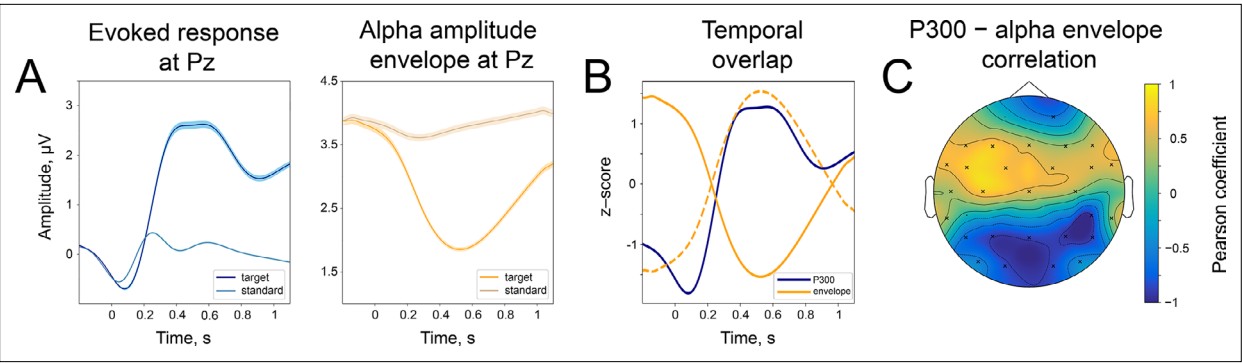

**Figure 2.** Temporal similarity between P300 and the alpha amplitude envelope. (**A**). Left panel—time course of P300 at the Pz electrode elicited by the target stimulus and ER after a standard stimulus (sER) both averaged across participants. Right panel—alpha amplitude envelope at Pz electrode averaged across participants for target and standard stimulus. Shaded areas display the standard error of the mean. Sample size is 2230. (**B**). Temporal overlap in signals. The time courses of P300 and alpha amplitude display similarities in initial slope and peak latency. Amplitude values are z-scores to aid visual comparison. Dashed line—alpha amplitude envelope multiplied by –1. (**C**). A correlation between P300 and alpha amplitude. For grand averages at each electrode, the correlation between P300 and alpha envelope was computed with the Pearson correlation coefficient. Electrodes marked with 'x' had significant correlation coefficients. The p-value was set at the Bonferroni corrected value of $10^{-4}$. Note the positive correlation between the low-frequency signal and the alpha amplitude envelope over central sites. Due to the negative polarity of ER over the fronto-central sites, such correlation may still indicate a temporal relationship between the P300 process and oscillatory amplitude envelope dynamics (due to the use of a common average reference). However, it cannot be entirely excluded that additional lateralised response-related activity contributes to this positive correlation (*Salisbury et al., 2001*).

The online version of this article includes the following figure supplement(s) for figure 2:

**Figure supplement 1.** Time-space evolution of P300 and alpha amplitude envelope.

a large EEG data set. In this data set, the experimental task was an auditory oddball paradigm. Participants would hear tones, one type of which—the target tone—would occur in only 12% of trials. Target tones elicit both P300 and the modulation of the alpha amplitude. Firstly, we show that in sensor space, the time courses of P300 and the alpha amplitude envelope are negatively correlated in the posterior region, and, in addition, the depth of alpha amplitude modulation correlates with the amplitude of P300. Secondly, we demonstrate that the increase in the low-frequency amplitude, that is P300, is pronounced over the posterior region, where at the same time the decrease in the alpha amplitude also occurs. Additionally, we perform source reconstruction to precise the location. Thirdly, by means of the baseline-shift index (BSI, *Nikulin et al., 2010*), we estimate oscillatory mean and establish that the sign of the mean is predictive of the P300-alpha relation. Finally, we evaluate the correlation between cognitive processes such as attention, memory, and executive function with P300 and alpha rhythm to confirm the relatedness of the two phenomena via behaviour.

## Results
### Temporal similarity between alpha amplitude envelope and P300

In line with the first prediction, average time courses of P300 and alpha amplitude envelope demonstrate an inverse relation—while P300 has a positive deflection, alpha rhythm amplitude is attenuated (*Figure 2*). The ER after the standard stimulus does not demonstrate the same strong relation (we will refer to ER after the standard stimulus as sER). To illustrate the relation even further, we filtered the ER in low frequency up to 3 Hz. *Figure 2A* on the left demonstrates the evolution of averaged time courses of ER at the Pz electrode, and *Figure 2A* on the right is the same but for the alpha amplitude envelope (see also *Figure 2—figure supplement 1* for the whole-head time courses). This figure clearly shows a similarity in the temporal evolution for both types of signals. More specifically, within a window 200–400ms after stimulus onset, P300 has a rising flank and alpha amplitude starts to decrease, and within a window 400–700ms, both P300 and alpha amplitude have the largest magnitude (*Figure 2B*). To quantify this relation, we estimated the correlation between P300 and alpha amplitude envelope over averaged signals at every electrode. As predicted, the correlation for the target stimulus was significantly negative at posterior regions (*Figure 2C*, at Pz correlation is –0.86).

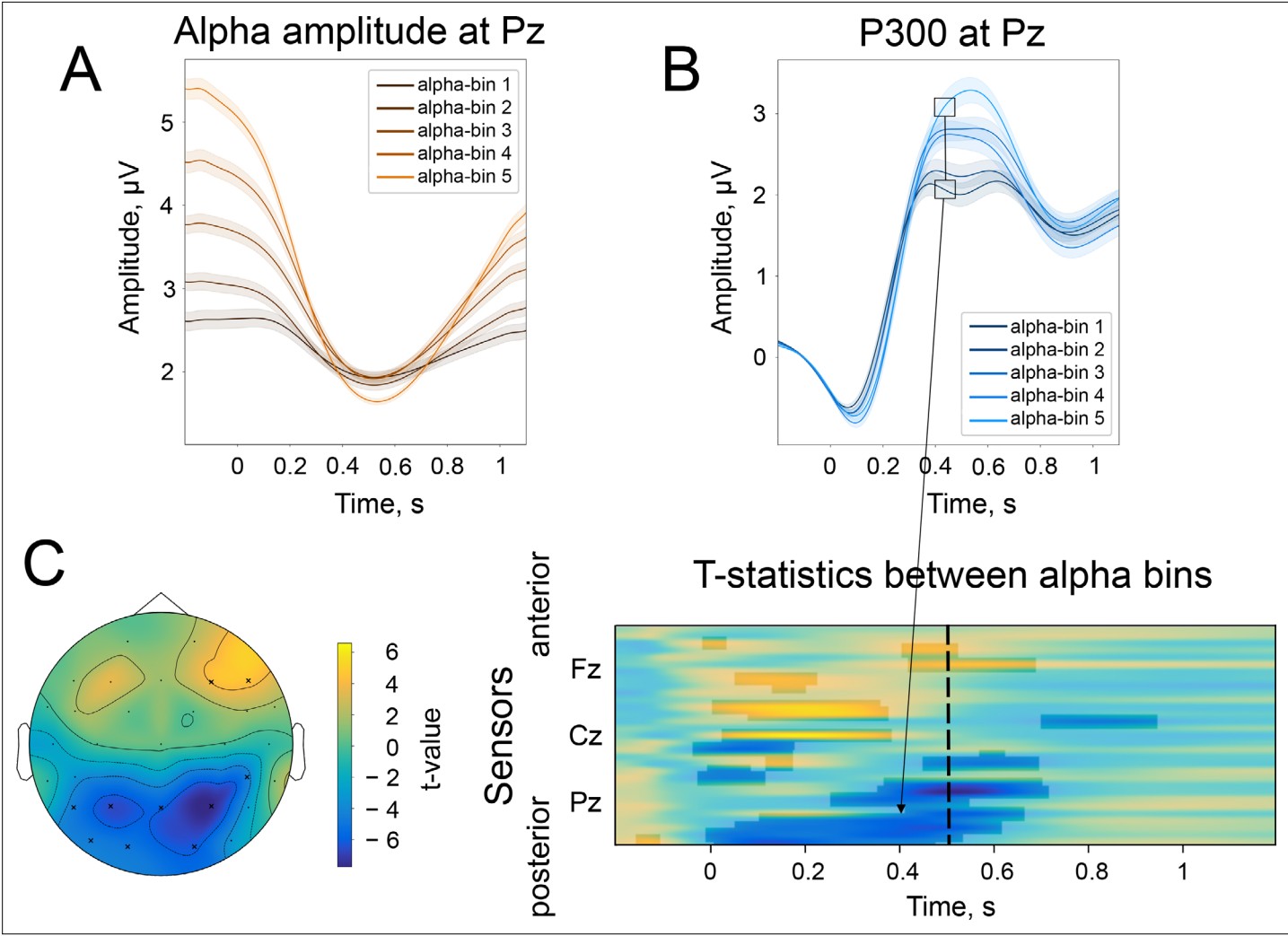

**Figure 3.** The difference in the strength of alpha amplitude modulation correlates with the difference in P300 amplitude. (**A**). Alpha amplitude envelope sorted into 5 bins according to the depth of modulation in the poststimulus window. The bins were the following: (66, –25), (–25,–37), (–37,–47), (–47,–58), (–58,–89)% change. Here, –100% corresponds to the deepest modulation, and 0% to the absence of a change in the amplitude. (**B**). P300 responses are sorted into the corresponding bins. Shaded areas display the standard error of the mean. Total sample size is 2230, sample size in each bin is 446. (**C**). The spatio-temporal t-test reveals clusters of significant differences between the two most extreme bins—bin 1 and bin 5. The topography of t-statistics is sampled at 500ms (dashed line). The significant electrodes at this time point are marked with "x".

The online version of this article includes the following figure supplement(s) for figure 3:

**Figure supplement 1.** The difference in the strength of alpha amplitude modulation correlates with the difference in early ER, but only for broadband data.

**Figure supplement 2.** The synchronisation in the population of neurons generating alpha rhythm affects the amplitude of the alpha rhythm but not the evoked response.

According to the baseline-shift mechanism, the change in the strength of the amplitude modulation should be mirrored in the change in P300 amplitude. Indeed, when we sorted alpha amplitude envelopes between participants into 5 bins according to the normalised change, the P300 amplitude followed the partition of alpha amplitudes. The normalised change was computed as $\frac{A_{post}-A_{pre}}{A_{pre}} * 100\%$, meaning that a value closer to –100% corresponds to a strong drop in the poststimulus amplitude in comparison to prestimulus, while a value closer to 0% corresponds to the absence of change in the amplitude, and a value larger than 0% corresponds to the increase in the amplitude in the poststimulus window. The different alpha amplitude dynamics correlated with P300 amplitude, such that for participants with a stronger alpha amplitude modulation, the amplitude of P300 was higher than for participants with weak amplitude modulation (*Figure 3A and B*). As predicted by BSM, a

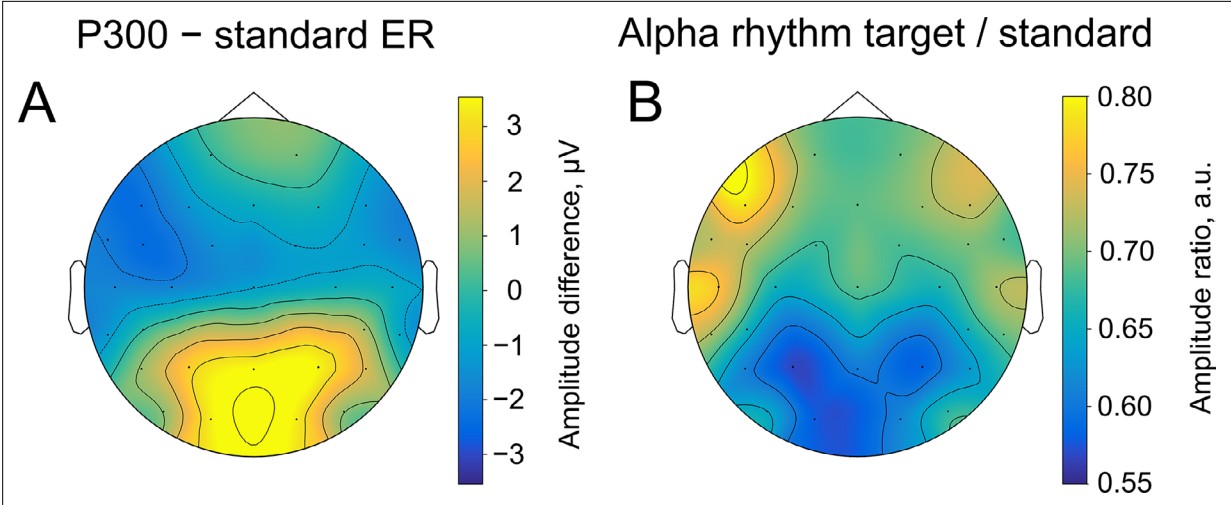

**Figure 4.** Spatial similarity of topographies of P300 (**A**) and alpha amplitude (**B**) contrasted between the target and standard stimulus. The topographies are shown at the peak amplitude of P300, which was estimated from the averaged over trials ER for each participant within the time window of 200–1000ms poststimulus at the Pz electrode (on average 509±171ms). For ER, the contrast was built by subtracting the sER amplitude from the P300 amplitude. For alpha amplitude, the contrast was built by dividing values of the amplitude after the target stimulus onto values after the standard stimulus.

smaller alpha amplitude modulation will generate an ER with a smaller amplitude. The total number of participants in each bin is 446. The t-test between the most extreme bins demonstrates a significant spatio-temporal cluster in the posterior region spanning electrodes CP6, P3, P4, P7, Pz, O1, O2, PO9 (*Figure 3C*). Here, t-values are negative, meaning that P300 that coincides with small alpha amplitude attenuation is significantly smaller in its amplitude than P300 that coincides with the largest alpha amplitude attenuation. The cluster within the earlier window (100–200ms) over central regions (*Figure 3C*) possibly reflects the previously shown effect of prestimulus alpha amplitude on earlier ERs (*Brandt et al., 1991*; *Babiloni et al., 2008*) but may also be a manifestation of BSM. We tested this assumption for early ER, which in our auditory task was N100. We repeated the binning analysis for broadband data (0.1–45 Hz) and also observed a significant difference between two extreme bins around 100ms over the central region (*Figure 3—figure supplement 1A*). However, if we filter the signal from 4 to 45 Hz (the range that includes the frequency of N100 but not low-frequency baseline shifts), these significant differences almost completely disappear (only electrode TP9 was significant; *Figure 3—figure supplement 1B*). It means that the difference in N100 amplitudes over frontal sites is driven by the baseline shift created by an unfolding alpha amplitude decrease. The significant difference at the TP9 electrode possibly reflects a genuine physiological effect of alpha rhythm amplitude on the excitability of a neuronal network and, as a consequence, on the amplitude of ER (as opposed to the baseline-shift mechanism, where the alpha rhythm does not affect the amplitude of ER but creates an additional component of ER; *Iemi et al., 2019*).

## Spatial similarity between alpha amplitude envelope and P300 in sensor space

Consistently with the second prediction, spatial distributions of P300 and alpha amplitude modulation overlapped considerably (Spearman correlation between topographies –0.80, p-value <0.0001, *Figure 4*). The highest amplitude of P300 (as contrasted with sER) is localised over posterior electrodes. Similarly, the highest alpha amplitude change (also contrasted with alpha amplitude after standard stimulus) appears in the same region. The topographies were sampled at the peak of P300, which on average happened at 509±171ms after the stimulus onset. The topography of ER (*Figure 4A*) was computed as the difference between the target and standard topography. The topography for alpha oscillations (*Figure 4B*) was computed as the ratio of amplitudes after the target and the standard stimuli. Note that the change in the alpha amplitude can be observed only through the contrast of target vs standard stimuli, since the topography of the target alpha amplitude retains prominent occipital alpha that may mask the reduction in the posterior region.

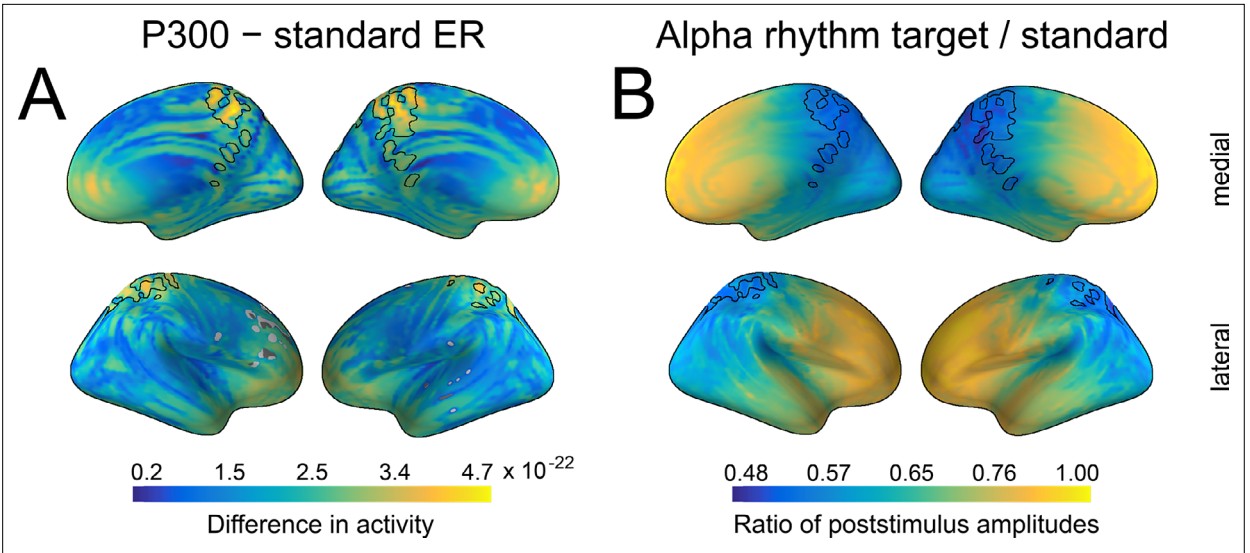

**Figure 5.** Spatial similarity between P300 and alpha amplitude in a source space. (**A**) The difference between P300 and sER, after correction for multiple comparisons. The difference was estimated as the subtraction of averaged sER power from averaged P300 power in the time window of 300–700ms. The colorbar thus indicates the difference in power. The black line outlines an overlap that is common for both P300 (top 10% of activity) and alpha amplitude (top 10% of activity). (**B**). The difference in alpha amplitude envelope after standard and target stimuli with a correction for multiple comparisons (all dipole locations are significant). The difference was estimated as the target poststimulus alpha amplitude divided by the standard alpha amplitude. The poststimulus window was the same as for P300: 300–700ms.

## Spatial similarity between alpha amplitude envelope and P300 in source space

To support the sensor space spatial similarity outcome and refine the spatial overlap location between P300 and alpha amplitude changes, we performed source reconstruction. As in sensor space, we juxtaposed activations from standard and target stimuli in source space, both for P300 and the change in alpha amplitude envelope. The biggest activations for both ER and alpha amplitude were localised on the parietal midline (precuneus, posterior cingulate cortex, BA 7, 31, 23; *Figure 5A and B*). The location of P300 is compatible with previous studies (*Tarkka et al., 1996*; *Tarkka and Stokic, 1998*; *Faro et al., 2019*) as well as with sensor space topography (*Figure 4*). For presentation, we outlined the overlap of dipole locations that was common for P300 and alpha amplitude change (the black line in *Figure 5A and B*).

## The decrease in alpha amplitude and positive deflection of P300 is explained by the sign of the oscillatory mean at resting state

In support of the third prediction, the sign of BSI, which determines the sign of the oscillatory mean, should also define the P300 polarity with respect to the alpha amplitude change. That is, for oscillations with a negative mean, the attenuation of amplitude will produce an ER with a positive polarity, whereas oscillations with a positive mean will lead to an ER with a negative polarity (see also *Figure 1* and *Video 1*). The BSIs for each participant at each electrode were estimated from a 10 min resting-state recording. The BSIs tended to be negative on average at Pz and in the nearby occipital region (*Figure 6A*). The distribution of BSIs at Pz was skewed towards negative values

Modulation of oscillations
with a negative
non-zero mean
generates
an evoked response
with positive polarity

**Video 1.** The demonstration of the baseline-shift mechanism for negative and positive non-zero mean oscillations that experience a stimulus-triggered increase or decrease in the amplitude.
https://elifesciences.org/articles/88367/figures#video1

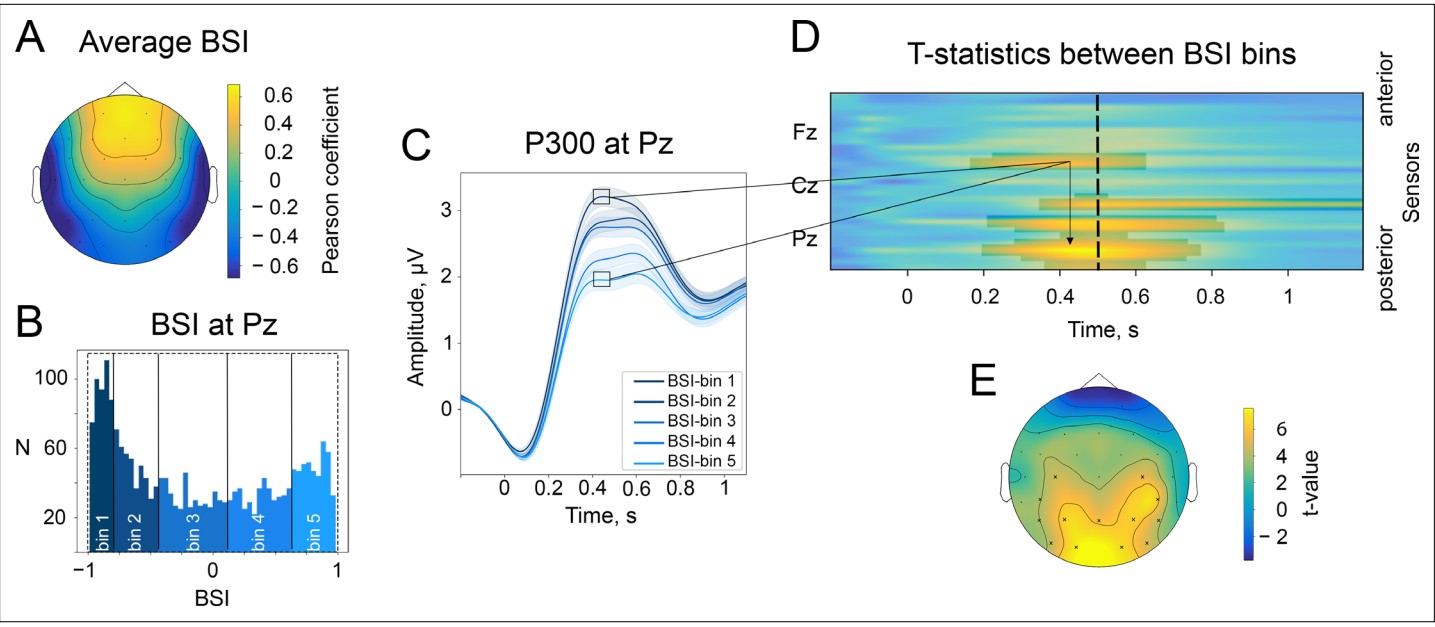

**Figure 6.** The baseline-shift index (BSI) explains the direction of ER based on the direction of alpha amplitude change. (**A**). The average values of BSI at each electrode estimated from the resting-state data. Here, BSI is computed as the Pearson correlation coefficient (see Methods/The baseline-shift index). BSI serves as a proxy for the relation between ER polarity and the direction of alpha amplitude change (**Nikulin et al., 2010**). Here, we observe predominantly negative BSIs (and thus negative mean oscillations) at posterior sites, which indicates the inverted relation between P300 and alpha amplitude change. Indeed, in the task data, a positive deflection of P300 at posterior sites coincides with a decrease in alpha amplitude. (**B**). BSIs at Pz were binned into 5 bins. The BSI bins were the following: (−0.99,−0.81), (−0.81,−0.46), (−0.46, 0.09), (0.09, 0.62), (0.62, 0.98). According to predictions of BSM, if BSI (and the oscillatory mean) was negative, then the attenuation of oscillations would lead to the upward direction of ER. (**C**). P300 was binned into bins according to BSI. For bins with negative BSI, the amplitude of P300 is higher in comparison to bins with positive BSI. Shaded areas display the standard error of the mean. Total sample size is 2230, sample size in each bin is 446. (**D**). The evolution of the statistical difference between the amplitude of P300 in the first and fifth BSI-bins across time and space. The difference is prominent over the central and parietal regions. The cluster-based permutation test revealed significant clusters in central and parietal regions with a p-value $10^{-4}$. (**E**). The topography of t-statistics is sampled at 500ms (at the dashed line of the upper panel). The significant electrodes at this time point are marked with 'x'.

(**Figure 6B**), with a mean value of −0.12 and a mode of −0.85. The distribution had a trough around zero, which indicates that oscillatory activity more often was a non-zero mean.

At the sensor level, BSI computed from resting-state EEG defines the changes in P300 according to BSM (**Figure 6B–E**). To estimate the connection between BSI derived from the resting-state recording and P300 features, we binned BSI values into 5 bins across participants. Thus, in the first bin, there were participants with more negative BSIs (446 participants) at the particular electrode, and in the fifth bin, there were participants with positive BSIs (also 446 participants). At Pz, the BSI covaried with P300 amplitude in a way that more negative BSIs corresponded to higher amplitudes of P300 in accordance with BSM, and more positive BSIs were associated with smaller amplitudes (**Figure 6C**). This trend is observed in other posterior and central electrodes (**Figure 6D**), and we estimated significant clusters spanning electrodes FC5, C3, C4, CP5, CP6, P3, P4, P7, P8, Pz, O1, O2, PO9, PO10, and the time window of maximal P300 amplitude, approximately 300–700ms (**Figure 6D**).

## Cognitive processes correlate with P300 and alpha amplitude modulation

Stimulus-based changes in brain signals are thought to reflect cognitive processes that are involved in the task. A simultaneous and congruent correlation of P300 and alpha rhythm to a particular cognitive score would be another evidence in favour of the relation between P300 and alpha oscillations. Moreover, if thus found, the correlation directions should correspond to the predictions according to BSM. Along with the EEG data, in the LIFE data set, a variety of cognitive tests were collected, including the Trail-making Test (TMT) A&B, Stroop test, and CERADplus neuropsychological test battery (**Loeffler et al., 2015**). From the cognitive tests, we extracted composite scores for attention, memory, and executive function (**Liem et al., 2017**, see Methods/Cognitive tests) and tested the correlation

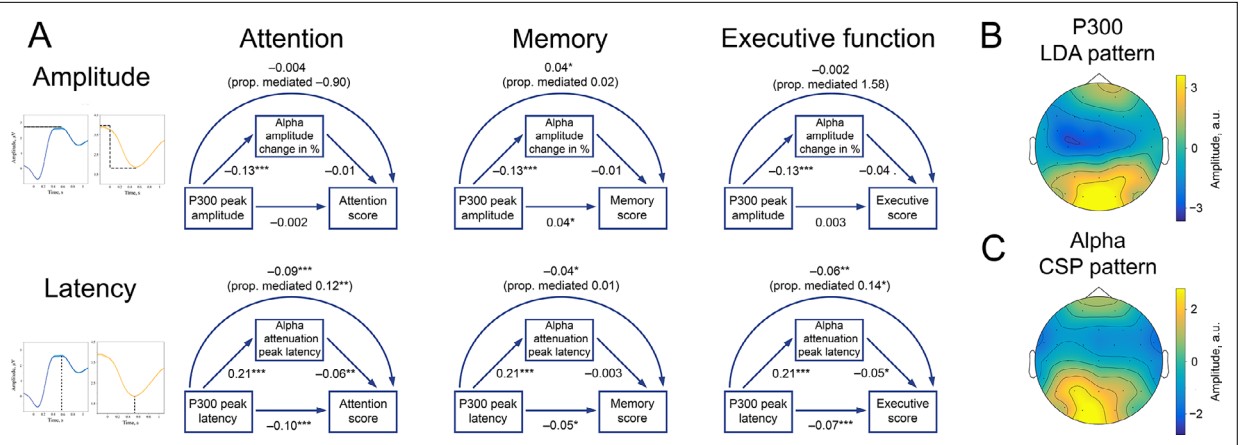

**Figure 7.** P300 and alpha oscillations showed similar correlation profiles across cognitive processes. (**A**). Attention, memory, and executive function scores correlate with P300 and the alpha envelope. Attention scores were computed from TMT-A time-to-complete and Stroop-neutral time-to-complete. Memory scores were computed from the CERAD word list (combined delayed recall, recognition, and figure delayed recall). Executive function scores were computed from TMT-B time-to-complete and Stroop-incongruent time-to-complete. P300 amplitude and latency were evaluated after spatial filtering with LDA. Alpha amplitude change and latency were evaluated after spatial filtering with CSP (see Methods/Spatial filtering). Beta values were estimated with linear regression having age as a covariate variable. Sample size for this analysis is 1549. ˙p-value <0.1, * p-value <0.05, ** p-value <0.01, *** p-value <0.001. Note that the alpha amplitude change direction is such that a lower negative value would correspond to a higher decrease. (**B**). A spatial pattern corresponding to the LDA spatial filter that was applied to obtain high signal-to-noise P300. (**C**). A spatial pattern corresponding to the CSP filter that was applied to obtain alpha oscillations.

between composite cognitive scores vs. P300 and vs. alpha amplitude modulation. The scores were available for a subset of 1549 participants (out of 2230), age range 60.03–80.01 years old. Cognitive scores correlated significantly with age (age and attention: −0.25, age and memory: −0.20, age and executive function: −0.23). Therefore, correlations between cognitive scores and electrophysiological variables were evaluated, regressing out the effect of age. To rule out the possibility of a absolute alpha power association with cognitive scores, for this analysis, we used alpha amplitude normalised change computed as $\frac{A_{post}-A_{pre}}{A_{pre}} * 100\%$, where $A_{post}$ is at the latency of strongest amplitude decsease. Computed this way, negative alpha amplitude change would correspond to a more pronounced decrease, that is stronger oscillatory response.

To increase the signal-to-noise ratio of both P300 and alpha rhythm, we performed spatial filtering (see Methods/Spatial filtering, *Figure 7B and C*). Following this procedure, both P300 and alpha latency, but not amplitude, significantly correlated with attention scores (*Figure 7A*, left column). Larger latencies were related to lower attentional scores, which corresponded to a longer time-to-complete of TMT and Stroop tests and hence poorer performance. The proportion of correlation between P300 latency and attention, mediated by alpha attenuation peak latency, is 0.12. Memory scores were positively related to P300 amplitude and negatively to P300 latency (*Figure 7A*, middle column). The direction of correlation is such that higher memory scores, which reflected more recalled items, corresponded to a higher P300 amplitude and an earlier P300 peak. The association between alpha rhythm parameters and memory scores is not significant, but it goes in the same direction as the association for P300. Executive function (*Figure 7A*, right column) were related significantly to both P300 and alpha amplitude latencies. The proportion of correlation between P300 latency and attention, mediated by alpha attenuation peak latency, is 0.14. Overall, the direction of correlation is similar for P300 and alpha oscillations, as expected for BSM. Moreover, the direction of correlation is consistent across cognitive functions.

## Discussion
### Generation of P300 is congruent with the baseline-shift mechanism
In the current study, we provided evidence for the hypothesis that the baseline-shift mechanism (BSM) is accountable for the generation of P300 to a certain extent. BSM for evoked response (ER) generation postulates that the modulation of oscillations with a non-zero mean leads to the generation of

ER (*Nikulin et al., 2007*; *Mazaheri and Jensen, 2008*). Here, we demonstrated the compliance of P300 generation with BSM using a large EEG data set. All the required prerequisites were confirmed: (1) the temporal courses of P300 and alpha amplitude were matching, (2) the spatial topographies of the P300 component and alpha oscillations were considerably overlapping, (3) the sign of the mean of alpha oscillations determined the direction of P300 given the decrease in alpha amplitude, (4) cognitive scores correlated in a similar way with the parameters of P300 and alpha amplitude. Therefore, P300, at least to some degree, is generated as a consequence of stimulus-triggered modulation of alpha oscillations with a non-zero mean.

The temporal correlation of P300 and alpha amplitude was negative in parietal regions. The amplitude of P300 was associated considerably with the prominence of alpha amplitude modulation, such that a smaller alpha amplitude modulation corresponded to a smaller P300 amplitude, and a larger, deeper modulation—to a larger P300. The significant cluster based on the spatio-temporal permutation test was also observed in parietal regions. Despite the fact that there is a distinct difference in P300 amplitude between participants who had a large and a small modulation, we would refrain from stating that a certain percent of P300 amplitude can be explained by alpha rhythm modulation. This conclusion cannot be definitive if we consider non-invasive recordings because spatial synchronisation within a population generating alpha rhythm greatly affects the scalp-level alpha amplitude but doesn't affect baseline shifts (see *Figure 3—figure supplement 2*).

The baseline-shift index (BSI, *Nikulin et al., 2010*) served as a method for estimating the mean of oscillations. The topographical distribution of the BSI differed from the P300 topography. However, because BSIs were estimated from resting-state recordings, they reflect the complex neurodynamics of various alpha-frequency sources, and resting-state BSIs cannot be expected to have the same topography as P300. Yet, BSI should be non-zero in the spatial locations similar to P300 and it should have a sign compatible with the generation of P300. This is indeed what we found: BSIs in the parietal region were mostly negative, which in correspondence with the direction of P300 in relation to alpha amplitude decrease (based on BSM). Furthermore, BSI was correlated with the amplitude of P300, with a significant relation between BSI and instantaneous ER amplitude observed in centro-parietal regions in a time window of 300–600ms after stimulus onset. The more negative BSI corresponded to higher amplitudes of P300. As posited by BSM, negative mean oscillations would generate an ER with a positive polarity.

Additionally, we tested the correlation of P300 and alpha rhythm with cognition. P300 is hypothesised to reflect attention, memory manipulation, and/or decision-making (*Polich, 2007*; *Verleger, 2020*), and previous studies showed that P300 correlated with attention (*Becker and Shapiro, 1980*; *Nakajima and Imamura, 2000*; *Lakey et al., 2011*), memory (*Watter et al., 2001*; *Braverman and Blum, 2003*; *Amin et al., 2015*), and executive function (*Kindermann et al., 2000*; *Dichter et al., 2006*). Alpha rhythm has been linked to attention (*Klimesch, 1999*; *Thut et al., 2006*; *Wislowska et al., 2022*) and memory (*Klimesch et al., 1997*; *Fellinger et al., 2012*; *van Ede, 2018*; *Wislowska et al., 2022*). In our study, scores reflecting attention, memory, and executive function have coincidental correlations to peak latency for both P300 and alpha oscillations. Namely, reduced attention and lower cognitive flexibility (*Kortte et al., 2002*; *Douw et al., 2016*) corresponded to increased peak latencies. Notably, the correlations of P300 and alpha rhythm with cognitive scores had a similar direction.

The mediation analysis showed that the modulation of alpha oscillations only partially explained the correlation between P300 and cognitive variables. This, in general, corresponds to the idea that not the whole P300 but only its fraction can be explained by the changes in the alpha amplitudes. *Figure 5* shows that alpha oscillations change not only in the cortical areas where P300 is generated; therefore, we cannot expect a complete correspondence between the two processes. Moreover, since cognitive tests and EEG recordings were performed at different time points, the associations between the cognitive variables and EEG markers are expected to be rather weak and to reflect only some neuronal processes common to P300, alpha rhythm, and tasks. For these reasons, a complete mediation of one EEG variable through another EEG variable in the context of a separate cognitive assessment cannot be expected.

## Previous reports on the concurrent alpha oscillations and P300

In our review of the previous literature (presented in the Introduction and Appendix 1), we found a large number of studies assessing simultaneously P300 and the oscillatory dynamics in the poststimulus

window. The majority of studies reveal the overlap in time windows and spatial regions of P300 and alpha amplitude decrease (see Appendix 1). However, not all the studies observed a complete overlap in time courses, especially as alpha rhythm remained suppressed beyond the P300 window. Moreover, there were studies that found some discrepancies between P300 and alpha oscillations. In one study (*Cooper et al., 2008*), the effects of TMS were observed only in alpha oscillations. Yet, the authors admitted that, possibly, the effect on P300 was not visible because the target of TMS—the right dorsolateral prefrontal cortex—did not include the P300 sources. For other studies that failed to find the relation (*Kamarajan et al., 2004*; *Tamura et al., 2016*; *Lee et al., 2017*; *Delval et al., 2018*), we hypothesise that the evidence of the link between P300 and alpha oscillations might have been obscured due to many alpha oscillations sources being present simultaneously (*Rodriguez-Larios et al., 2022*). Due to multiple alpha sources active at the same time, it is challenging to recover the exact alpha source that was responsible for ER generation. In particular, we observed high amplitude alpha oscillations in the occipital region (which is expected since participants were seated with their eyes closed). Moreover, the target tone presentation required participants to press the button, and as with any movement, the button press was also accompanied by oscillatory changes in the alpha (mu) frequency range (*Pfurtscheller and Lopes da Silva, 1999*; *Nikulin et al., 2008*). In line with this assumption, we found a positive correlation between ER and alpha amplitude envelope around C3-C4 electrodes (*Figure 2C*) and negative ER amplitudes over the same region (*Figure 4A*; also see *Figure 2—figure supplement 1A* where the P300 time courses have negativity over central electrodes), which indicates that, possibly, there is a motor-related component of ER (*Salisbury et al., 2001*), with typically observed negative polarity that may have originated from a source of alpha (mu) oscillations relating to motor activity. Hence, depending on the task, there might be other changes in rhythmic activity that occlude or completely hinder the identification of oscillations that are related to the ER in question. Furthermore, none of those studies explicitly tested the compliance of the P300 generation with BSM. In our study, we extended the analysis by showing that in the same brain region, resting-state baseline shifts related to the amplitude of the non-zero mean alpha oscillations in a similar way as P300 related to stimulus-triggered alpha amplitude change. It is important to note that when assessing the interrelatedness of ER and oscillatory processes via BSM, it is necessary to evaluate all BSM predictions.

## Alternative explanations

Previously, P300 origins have been assessed according to the predictions of the additive mechanism (*Fell et al., 2004*; *Wan et al., 2009*; *Herrmann et al., 2014*) and the phase-reset mechanism (*Fell et al., 2004*; *Daly et al., 2009*; *Wan et al., 2009*). Both mechanisms have been extensively researched for different ERs, but the assessment of P300 compliance with these mechanisms is rather problematic, as is the case in general for all non-invasive measures trying to disambiguate mechanisms of ER generation (*Telenczuk et al., 2010*). First, the additive mechanism postulates that ER is added to the overall activity (*Wood and Allison, 1981*; *Jervis et al., 1983*; *Mäkinen et al., 2005*). Consequently, ER should be accompanied by an increase in total power and not only oscillatory power. However, the P300 is always accompanied by an increase in low-frequency power in the theta range, as it is its frequency range. Therefore, the predicted increase in power exclusively due to the addition of activity (*Shah et al., 2004*; *Mazaheri and Jensen, 2006*) is impossible to disentangle based on macroscopic recordings (*Telenczuk et al., 2010*) and multi-unit activity is required to confirm whether an increase in power in the P300 window is of an oscillatory or non-oscillatory nature. Moreover, in fact, BSM can also mimic the evidence for the additive mechanism, such that an ER that is generated via BSM will always be accompanied by a change in power in the low-frequency range (*Figure 1*). Second, the phase-reset mechanism states that ER is created when a stimulus triggers the phase alignment of oscillators in a certain frequency (*Sayers et al., 1974*; *Makeig et al., 2002*; *Hanslmayr et al., 2007*). Analogously, due to the frequency content of P300, there would be increased phase consistency in the theta range since ER, be it of additive or phase-reset nature, always has phase alignment. Yet, phase alignment in the poststimulus window does not contradict BSM either. We argue that the current set of predictions for the additive and the phase-reset mechanism is insufficient to confirm the generation of P300 and needs further development. As for BSM, all four BSM prerequisites (verifiable with non-invasive EEG recordings) were validated in our study, and therefore it seems reasonable to conclude that the generation of P300 is congruent with the BSM model.

The evidence presented in the current study speaks for a partial rather than an exhausting explanation of P300's origin through BSM. The P300 is not a single ER but rather a complex. Previously (*Polich, 2003*; *Linden, 2005*), P300 was subdivided into the complex that has an earlier component—P3a—that occurs around 300ms after stimulus onset and is more prominent in the anterior midline, and a later component—P3b—that has a latency of 500ms and beyond and is present to a large extent in the parietal electrodes. Besides, with PCA decomposition of P300, several other components have been observed, namely slow wave and very late negativity (*Steiner et al., 2014*), which further indicates the complexity of the brain's response to a target stimulus. The known and investigated mechanisms of ER generation—additive mechanism, phase-resetting mechanism, and BSM—may explain different temporal windows of one ER (*Iemi et al., 2019*). In our research, we found that a slow low-frequency wave of P300 may be explained by the concurrent changes in alpha amplitude via BSM. It is nonetheless feasible that a certain part of P300 might still be generated via the additive or phase-reset mechanism, although, in contrast to BSM, the prerequisites for these two types of mechanisms are hard to verify with EEG/MEG (*Telenczuk et al., 2010*). Moreover, determining a certain variance of P300 amplitude that can be explained by BSM is challenging when we analyse non-invasive recordings. The synchronisation within a population generating alpha rhythm affects the scalp-level alpha amplitude (see *Figure 3—figure supplement 2*) such that for a poorly synchronised network, the power of oscillations is severely diminished (*Studenova et al., 2022*). However, since the baseline shift doesn't depend on the phase of oscillations, its amplitude is not influenced by the strength of synchronisation. Therefore, here, we did not aim to completely explain the P300 complex but to show that all four prerequisites for BSM are met for P300 generation, thus mechanistically linking P300 and alpha oscillations.

## Limitations

In our previous study (*Studenova et al., 2022*), using a smaller data set, we found that baseline shifts were harder to detect in the elderly population compared to the younger population. However, in the current study, due to a large sample size, we overcame difficulties related to the extraction of baseline shifts in aged participants and revealed statistically significant associations between ERs and oscillations. Essentially, the alpha amplitude decrease, triggered by the target stimulus, was particularly prominent and was substantial in the majority of participants. Only for 3% of participants, the amplitude of alpha rhythm after the target stimulus was equal to or greater than after the standard stimulus within the P300 window. In all other participants, a target stimulus evoked a pronounced attenuation of alpha oscillations. Besides, P300 in the elderly and patients with cognitive decline had smaller amplitude and longer latency (*van Dinteren et al., 2014*), but it never completely disappeared. In our sample, only 9% of participants had P300 amplitude smaller than sER. This in turn gave us an ample opportunity to investigate P300 and related alpha oscillations. The matter may be more complicated with other ERs that, for instance, are associated with the alpha rhythm that is generated by a smaller population of neurons and hence may be masked by other alpha rhythm sources with a higher amplitude, for example for auditory responses (*Weisz et al., 2011*).

A noteworthy limitation of the study is that EEG data was collected using only 31 channels. Spatial mixing is a substantial problem for any EEG set-up, and a smaller number of electrodes complicates the oscillatory analysis further. It was shown that with a small number of electrodes, the spatial accuracy of source reconstruction deteriorates (*Liu et al., 2018*; *Dattola, 2020*). In our case, in the oddball paradigm, both P300 and alpha amplitude changes were clearly detectable, and we expected that the corresponding ROIs would be rather large, and thus 31 electrode coverage would be sufficient. For other paradigms or other frequencies (like beta and gamma), the resolution of 31 channels may be insufficient since these rhythms are generated by a smaller number of neurons (*Pfurtscheller and Lopes da Silva, 1999*).

## Implications of P300 and alpha rhythm relation

The detected link between P300 and alpha oscillations provides a novel avenue for the P300 interpretation, as the P300 functional role remains a subject of active discussion (*Polich, 2007*; *Verleger, 2020*). It has been suggested that P300 corresponds to the inhibition of irrelevant activity, which is needed to facilitate the processing of a stimulus or task (*Polich, 2007*). However, because a decrease in alpha rhythm amplitude is considered an indication of disinhibition of a particular

region (*Pfurtscheller and Lopes da Silva, 1999*; *Jensen and Mazaheri, 2010*), it would follow that P300 may rather act as a correlate of activation related to the processing of the target stimulus. In previous research, alpha has been associated with attention (*Foxe and Snyder, 2011*; *Klimesch, 2012*; *Peylo et al., 2021*), and working memory (*Freunberger et al., 2011*; *de Vries et al., 2020*). Attentional processes were reflected in the changes in alpha amplitude, such that it increased to suppress distractions and decreased to facilitate relevant processes (*Neuper and Pfurtscheller, 2001*; *Van Diepen et al., 2019*), while associations with working memory demonstrated inconsistent amplitude changes (*Rodriguez-Larios et al., 2022*). Therefore, we propose that at least partially, P300 reflects the disinhibition of regions responsible for attention and, possibly, working memory.

Here, we investigated the role of alpha oscillations in the generation of the P300 evoked response. However, our analytic pipeline may be easily applicable to any other ER that usually coincides with the modulation of any oscillations (in the form of a decrease or increase in the amplitude). The ERs suitable for testing against predictions of BSM include contingent negative variation (CNV), N400, earlier left anterior negativity (ELAN) and readiness potential (as they coincide with oscillatory changes in the alpha range, see *Filipović et al., 2001*; *Bastiaansen et al., 2002*; *Bender et al., 2004*; *Shibasaki and Hallett, 2006*; *Heimann et al., 2017*). This list is not complete and may include other ERs and oscillations of higher or lower frequencies.

In the current study, we found that the attenuation of alpha amplitude in parietal regions gives rise to the slow component of positive polarity in the P300 complex. Although sometimes analysed together, previously, P300 and alpha rhythm were not considered to represent the same neuronal process. We, on the other hand, demonstrated that alpha oscillations, at least partially, give rise to a P300 via the baseline-shift mechanism. Based on the results of our study, we suggest that general inferences about changes in P300 amplitude or latency should be derived in conjunction with changes in oscillatory dynamics. Overall, we provide a framework and evidence for the unifying mechanism responsible for the generation of ERs from amplitude dynamics of neuronal oscillations.

## Methods

### Key resources table

| Reagent type (species) or resource | Designation | Source or reference | Identifiers | Additional information |
|---|---|---|---|---|
| Software, algorithm | MNE-Python | *Gramfort et al., 2013* | | |
| Software, algorithm | Python autoreject | *Jas et al., 2016*; *Jas et al., 2017* | | |
| Software, algorithm | Python scipy | *Jones et al., 2001* | | |
| Software, algorithm | Python sklearn.discriminant_analysis | *Pedregosa et al., 2011* | | |
| Software, algorithm | R lme4 | *Bates et al., 2015* | | |
| Software, algorithm | R mediation | *Tingley et al., 2014* | | |

### Participants

The LIFE data set (*Loeffler et al., 2015*) contains data from approximately 10,000 individuals aged 40–79 years. All participants gave their written informed consent. For our study, we selected participants who took part both in resting and stimulus EEG sessions (a total of 2886 participants). From that, we had to remove 12 due to inconsistencies in stimuli coding and the mismatched header files, and 7 due to short recordings. We included all participants with no obvious neurological and psychological disorders at the moment of testing (97 participants were rejected due to medications taken at the time of data collection). We assessed the quality of the data by checking the electrode-level spectra of both resting and stimulus-based recordings. Based on the visual inspection of the quality of spectra in the low-frequency range (significant noise in more than two channels, noise in a low-frequency range of larger amplitude than the alpha peak), we rejected 539 participants (451 based on resting-state recordings, 282 based on stimulus recordings, some of them overlap). The resulting sample contained 2230 participants, aged 60–82 years old, 1152 females.

## Resting session

During the day of the recording, each participant went through three sessions: resting-state session, the oddball-novelty stimuli session, and the intensity dependence of acoustically evoked potentials session. The total time of EEG recording with preparation and follow-up did not exceed 120 min. The EEG resting session was recorded for a total of 20 min with an eyes-closed state. Thirty-one electrodes were used for the recording, with additional electrodes for vertical and horizontal eye movements and heartbeats (40-channel QuickAmp amplifier). The electrode positions were already fastened on the cap according to the international 10–20 system. The impedances were kept under 10 kOm. The data were sampled at 1000 Hz with a low-pass filter at 280 Hz. The recording was performed with the common average reference (*Jawinski et al., 2017*). Before the EEG resting session, participants were situated in a reclined position, and instructed to relax and not to resist the urge to fall asleep. Based on the predictions of BSM, from the resting-state signal, we derived the association between alpha amplitude and corresponding low-frequency baseline shifts and quantified it with the baseline-shift index (BSI, see Methods/The baseline-shift index). To compute BSI, we assessed only the first 10 min after the beginning of the recording to decrease the possibility of the participants falling asleep.

## Oddball session

P300 was assessed by employing an acoustic oddball paradigm with three stimuli: standard, target, and novelty. A hearing test was carried out before the stimulus session to determine the hearing threshold for standard and target experimental stimuli. The hearing threshold was adjusted separately for each ear. Additionally, before the main experiment, a short test session was conducted to familiarise participants with standard and target stimuli and to make sure that they understood the instructions correctly. The main experimental session continued for 15 min; within that time, a total of 600 stimuli were presented in a pseudo-randomised order. At least two standard stimuli occurred between the target stimuli and no more than nine standard stimuli occurred in succession. The inter-stimulus interval was invariable and set to 1500ms. The standard (more frequent) stimulus appeared with a probability of 76%. Non-frequent stimuli, target and novelty, appeared with a 12% probability each. The standard stimulus was a sinusoidal tone with a frequency of 500 Hz, an intensity of 80 dB, and a duration of 40ms (including a 10ms rise and fall flanks). The target tone had the same characteristics as the standard, except for frequency, which was set to 1000 Hz for targets. The novelty stimuli were environmental or animal sounds, with an intensity of 80 dB and an average duration of 400ms (the rise and fall flanks were selected depending on the type of tone). Participants were instructed to press the button when they heard the target stimulus. After 300 stimuli, participants had a 30 s break. During the break, participants were asked to change the hand used to make the response (the hand was randomly assigned at the beginning of the experimental session). In this work, we focus on the ER after a target stimulus, using the ER after a standard stimulus as a contrast condition.

## Cognitive tests

Along with the EEG data, the LIFE data set also includes a large number of cognitive tests (*Loeffler et al., 2015*). To test the correlation of P300 and alpha oscillations with cognition, we selected tests that evaluated attention, memory, and executive function—cognitive processes that, in previous research, were shown to correlate with P300 and alpha rhythm (*Nakajima and Imamura, 2000*; *Lakey et al., 2011*; *Amin et al., 2015*; *Dichter et al., 2006*; *Klimesch, 1999*; *Thut et al., 2006*; *Fellinger et al., 2012*).

Attention scores were computed from the Trail-making test (TMT) and Stroop test (*Liem et al., 2017*). TMT is a neuropsychological test that usually includes two tasks (*Reitan, 1992*). The first task, also referred to as TMT-A, requires a participant to connect numbers from 1 to 25 in ascending order as quickly as possible. The second task, also referred to as TMT-B, introduces letters in addition to numbers and requires to connect both letters and numbers in an alternating fashion in ascending order. The Stroop test was performed as a computer-based colour-word interference task (*Zysset et al., 2001*; *Scarpina and Tagini, 2017*) with two conditions—neutral and incongruent. For attentional correlates, we selected the time-to-complete metric from TMT-A and time-to-complete in the neutral condition from the Stroop test (*Kynast et al., 2018*; *Treviño et al., 2021*). In each test, not-a-number values and implausible answers were filled with the mean values of the rest of the sample. After that,

both metrics were standardised with z-score (sklearn.preprocessing.StandardScaler, *Pedregosa et al., 2011*) and inverted (1/value). The average of two values was taken as a composite attentional score.

Memory scores were derived from the CERADplus test battery. The CERAD neuropsychological test battery was developed by the Consortium to Establish a Registry for Alzheimer's Disease. In the LIFE data set, an authorised German version was used (https://www.memoryclinic.ch/de/, *Morris et al., 1988*; *Morris, 1989*). From the CERAD panel, we selected the delayed word recall score, delayed word recognition score, and delayed figure recall score. Every score represented the number of correctly recalled or recognized words or figures divided by the total number of possible correct answers (thus the scores were in the range from 0 to 1). Deviations from the normal answers, such as a refusal to answer, were set to zero. Lastly, each score was standardised (z-transformed) and an average of three values was taken as a composite memory score.

Executive function scores were compiled using the TMT and Stroop tests (*Liem et al., 2017*). From TMT, we took time-to-complete in condition B, and from the Stroop test, time-to-complete in the incongruent condition. The composite executive function score of each participant was an average of two scores—standardised (z-score) inverted TMT-B time-to-complete and standardised (z-score) inverted Stroop-incongruent time-to-complete.

## Preprocessing of resting-state and stimulus-based EEG data

The preprocessing of EEG data was performed with the MNE-Python package (*Gramfort et al., 2013*). For each participant, we preprocessed resting-state EEG in the following way. After loading the data, we performed re-referencing to an average common reference. Then, we filtered the recording in a wide bandpass range, from 0.1 Hz to 45 Hz, with the addition of a notch filter around 50 Hz. Bad channels and bad segments were removed based on a visual inspection (*Cesnaite et al., 2023*) and based on markers set by the recording technician. Additionally, we visually verified the spectrum of each participant's EEG for noisy channels. For further analysis, we exported the first ten minutes of resting-state recording. Using this time window, we computed BSI at each electrode (see Methods/ The baseline-shift index).

Next, we pre-processed EEG data from a stimulus-based oddball paradigm. We applied average reference to the data and filtered it in a range from 0.1 Hz to 45 Hz, with the addition of a notch filter around 50 Hz. Continuous data were cut to trials of 1.7 s long, starting at −0.4 s before stimulus onset, and baseline corrected, with the baseline taken as −0.2,−0.05 s before stimulus onset. We applied trial rejection based on markers set by the recording technician and based on high amplitude artefacts (detected with autoreject; *Jas et al., 2016*; *Jas et al., 2017*), then we used automatic Python functions to detect eye artefacts (mne.preprocessing.create_eog_epochs) and dampened them with signal space projection (SSP, *Uusitalo and Ilmoniemi, 1997*), discarding one single component (conservative choice to preserve the signal of interest). To compute the single-trial ER, we low-pass filter the data using the Python scipy.signal module (*Jones et al., 2001*) at 3 Hz. To compute single-trial changes in alpha amplitude, we subtracted the average broadband ER from each trial, then band-pass filtered each trial around the individual alpha peak for each sensor (estimated with the spectrum obtained by Welch's method, scipy.signal.welch) and extracted the alpha amplitude envelope with the Hilbert transform (scipy.signal.hilbert). Lastly, we averaged both the ER and the alpha amplitude envelope over trials (*Figure 1*).

## The baseline-shift index

The baseline-shift index (BSI, *Nikulin et al., 2010*) estimates a non-zero mean of oscillations based on the fluctuations in their amplitude, as proposed by the baseline-shift mechanism (BSM, *Figure 8*). BSM can be summarised with the equation:

$$y\left(t\right) = A\left(t\right)\left[cos\left(2\pi ft + \theta\right) + r\right] = A\left(t\right)cos\left(2\pi ft + \theta\right) + A\left(t\right)r \tag{1}$$

where $y\left(t\right)$ —data from a single oscillator or a coherent population of oscillators,

$f$ —some arbitrary frequency of oscillations in the population,

$\theta$ —some arbitrary phase,

$r$ —non-zero oscillatory mean,

$A\left(t\right)$ —amplitude modulation,

$A\left(t\right)r$ —a baseline shift that accompanies oscillations.

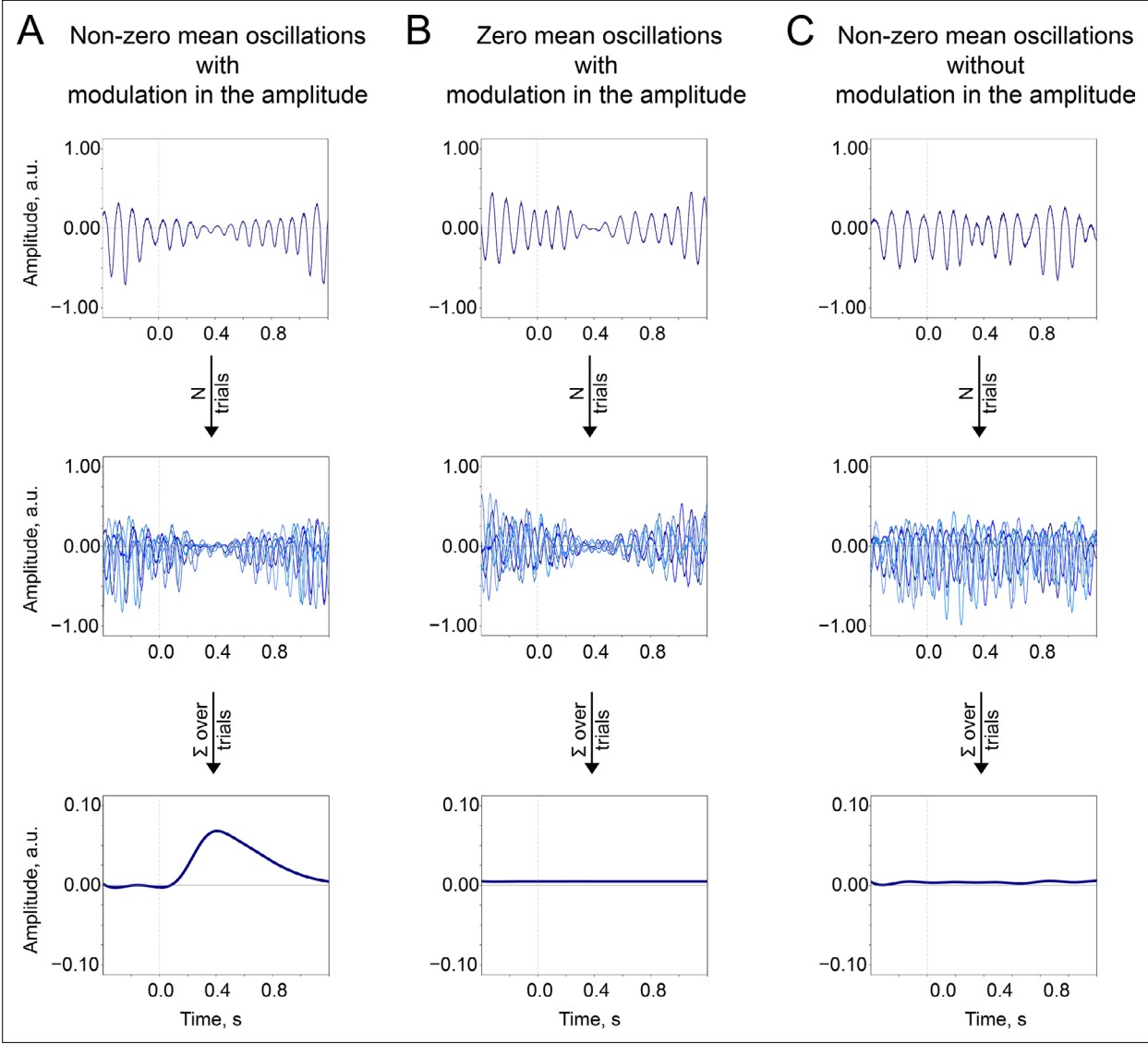

**Figure 8.** The baseline-shift mechanism (BSM) summary. Two important prerequisites of the BSM—non-zero mean $r$ and amplitude modulation $A\left(t\right)$—should occur together so the ER would be generated. (**A**). Non-zero mean oscillations when modulated in amplitude generate an ER. (**B**). If oscillations have a zero mean, then no ER is generated. (**C**). If oscillations have a non-zero mean but do not systematically (trial-by-trial) experience modulation, then no ER is generated.

Building on the predictions of BSM, in empirical EEG recordings, non-zero mean oscillations leave a 'trace' in the low-frequency range. Therefore, the evidence of non-zero mean property for oscillations in question can be accumulated by measuring a correlation between modulation of oscillations' amplitude (in the form of an amplitude envelope, $A\left(t\right)$ in *Equation 1*) and low-frequency signal (that presumably contains baseline shifts, $A\left(t\right)r$ in *Equation 1*). Note that the computation of BSI should be carried out using the resting-state recording to avoid contamination by stimulus effects.

A detailed description can be found in previous works (*Nikulin et al., 2010*; *Studenova et al., 2022*). In brief, firstly, we created two signals (1) by filtering broadband data in the alpha band (+−2 Hz around individual alpha peak frequency) and (2) by filtering original broadband data in the low-frequency band (low-pass at 3 Hz). Filtering in the alpha band was performed with a zero-phase Butterworth filter of fourth-order, and filtering of a low-frequency signal—with a zero-phase Butterworth filter of eighth-order (scipy.signal.butter, scipy.signal.filtfilt). From filtered alpha oscillations, we derived an amplitude envelope using the Hilbert transform. Secondly, we binned alpha amplitude into 20 bins, from the smallest to the biggest amplitude. Using the same allocation, we placed a low-frequency signal into

bins as well. Amplitude values inside each bin were averaged, thus creating 20 corresponding points for alpha amplitude and low-frequency amplitude. Lastly, the relation between alpha amplitude and low-frequency amplitude was estimated as the Pearson correlation coefficient.

## The temporal and spatial similarity of alpha amplitude and P300

P300 appears in response to the target but not to the standard stimulus. Similarly, prominent alpha modulation occurs after target stimulus presentation in comparison to standard stimulus. To quantify the relation between both processes, we compared the topographical distribution of P300 and alpha amplitude dynamics in the poststimulus window around the peak of P300. We detected the peak amplitude of P300 from a filtered averaged ER in a window of 200–1000ms at the Pz electrode. Fifty-seven participants (only 2.6% of the total number of participants) did not have an identifiable peak; those participants' topographies were fixed at 500ms. The topography of ER was computed as the difference between target and standard topography. The topography for alpha oscillations was computed as the ratio of amplitudes after the target and the standard stimuli. The poststimulus window for alpha amplitudes was chosen according to the ER peak latency as $(t_{peak} - 50, t_{peak} + 50)$ ms. In the source space, the difference in evoked activations was estimated as the subtraction of averaged sER power from averaged P300 power in the time window of 300–700ms. For the alpha amplitude envelope, the difference was estimated as the target amplitude divided by the standard amplitude in the poststimulus window 300–700ms.

## Spatial filtering

While the averaged P300 and alpha amplitude envelope in a sensor space had distinctive similarities, not all of the participants had a high signal-to-noise ratio time course. To obtain a clearer time course estimate for each participant, we performed spatial filtering. For P300 derivation, we applied Linear Discriminant Analysis (LDA; *Blankertz et al., 2011*, sklearn.discriminant_analysis.LinearDiscriminantAnalysis, *Pedregosa et al., 2011*) obtained over all participants. To achieve that, first, we computed averaged time courses of P300 and sER (ER after standard stimulus) for each participant. Second, we obtained averaged amplitude in the time window from 300 to 700ms, thus creating two values of amplitude for each participant. Third, the LDA was trained with P300 amplitude values (matrix—participants by electrodes) and sER amplitude values (matrix—participants by electrodes) as two distinct classes. The result is the spatial filter, which is a set of weights for electrodes that maximise the difference between classes while minimising the variance inside the class, thus providing the largest discriminability between the classes. The spatial pattern was derived with the spatial filter and covariance of the cumulative data using the formula $A = \frac{(Cov, W)}{((W^T, Cov), W)}$ , where $A$—spatial pattern, $W$—spatial filter, $Cov$—covariance of stacked data (*Schaworonkow and Nikulin, 2022*). Lastly, the weights derived from LDA were applied to the data of every participant, thus obtaining a single time course of P300. From this time course, the peak amplitude and peak latency were extracted for further analysis.

For alpha oscillations, we applied a CSP spatial filter (code based on *Schaworonkow and Nikulin, 2022*, scipy.linalg.eig). First, we obtained covariance matrices for each participant for each condition, averaged over trials. Specifically, for each participant, we computed a covariance matrix of every alpha-filtered trial in the time window from 300 to 700ms and then averaged trial-based matrices within the condition (*Zuure and Cohen, 2021*). Second, we averaged covariance matrices to obtain a grand average over the sample of all participants. Third, using two covariance matrices of target and standard stimuli, we computed CSP filters and corresponding patterns. From those patterns, we selected the one that had the largest similarity to the P300 topography. This was the first CSP component with the largest eigenvalue. Lastly, we applied a selected filter to the data of every participant. Then the spatially filtered alpha oscillations from every trial were processed with Hilbert-transform to compute the amplitude envelope, as before (see Methods/Preprocessing of resting-state and stimulus-based EEG data). From the averaged-over-trials amplitude envelope, we derived the latency and the amplitude of the attenuation peak.

## Source reconstruction

After initial preprocessing, the stimulus-based data were filtered in the band 0.1–20 Hz and decimated to the sampling rate of 100 Hz and all trials have been concatenated for further source reconstruction. We used source localisation based on the fsaverage subject (Python module mne.

minimum_norm, mne-fsaverage) from FreeSurfer (*Fischl, 2012*). A 3-layer Boundary Element Method (BEM) model was used to compute the forward model. Source reconstruction was carried out with eLORETA (*Barry et al., 2020*) with the following parameters: free-orientation inverse operator (loose = 1.0), normal to the cortical surface orientation of dipoles, the regularisation parameter lambda = 0.05, and the noise covariance is the covariance of white noise signals with equal duration to the data (*Idaji et al., 2022*). Source spaces had 4098 candidate dipole locations per hemisphere. Reconstructed data were split into trials after reconstruction and passed through the processing pipeline as in sensor space, namely, averaging ER and computing alpha amplitude envelope (*Figure 1*).

## Statistical evaluation

To estimate the statistical significance for sensor-space data, if not mentioned otherwise, we used Bonferroni corrected p-values. Namely, in a sensor space, the threshold for each electrode was set as a p-value = $10^{-4}/31$ . For source space, the threshold was chosen in a similar fashion: p-value = $10^{-4}/8196$ .

In source space, we identified an overlap of the most prominent activity. For each dipole location on a cortical surface mesh, we computed t-statistics between the amplitude of sER and P300 in a time window from 300 to 700ms. From all locations that have a significant difference, we took those that have the biggest difference in power between P300 and sER (top 10%). Similarly for alpha amplitude, we identified all significant locations, and then took 10% of those significant locations that had the biggest ratio of target to standard alpha amplitude in the poststimulus window of 300–700ms. Then, we extracted the overlap between the two regions of interest for presentation purposes.

We ran additional statistical analysis to test the relation between P300 amplitude and the depth of alpha amplitude modulation in a sensor space. The depth of amplitude modulation was computed as $\frac{A_{post} - A_{pre}}{A_{pre}} * 100\%$. For each electrode, we binned the alpha amplitude modulation of all participants into 5 bins. Then, we used this binning to sort P300. Next, for each time point, we computed t-statistics between the amplitude of P300 at that point in the 1st and 5th bins (which corresponded to the smallest and the largest modulation respectively). To evaluate significance, we ran a cluster-based permutation test (Python mne.stats.spatio_temporal_cluster_test) with 10,000 permutations and the threshold corresponding to a p-value = $10^{-4}$ .

A separate statistical test was carried out for the relation of P300 amplitude with the BSI in sensor space. For each electrode, we binned the ERs of all participants into 5 bins according to the value of BSI at that particular electrode in the resting-state recording. For each time point, we computed t-statistics between the amplitude of P300 in the 1st and 5th BSI bins (which corresponded to the most negative BSIs and the most positive BSIs for this particular electrode). After that, we ran a cluster-based permutation test (Python mne.stats.spatio_temporal_cluster_test) with 10000 permutations and the threshold corresponding to a p-value = $10^{-4}$ .

The correlation between cognitive scores (see Methods/Cognitive tests) and the amplitude and latency of P300 and alpha oscillations was calculated with linear regression using age as a covariate (R lme4, *Bates et al., 2015*). To estimate what proportion of the correlation between P300 and cognitive score is mediated by alpha oscillations, we used mediation analysis (*Baron and Kenny, 1986*; R mediation, *Tingley et al., 2014*). First, we estimated the effect of P300 on the cognitive variable of interest (total effect, *cogscore ~ P300 + age*). Second, we computed the association between P300 and alpha oscillations (the effect on the mediator, *alpha ~ P300*). Third, we run the full model (the effect of the mediator on the variable of interest, *cogscore ~ P300 + alpha + age*). Lastly, we estimated the proportion mediated.

# Additional information

### Competing interests
Denis Alexander Engemann: D.E. is a full-time employee of F. Hoffmann-La Roche Ltd. The other authors declare that no competing interests exist.

## Funding

| Funder | Grant reference number | Author |
|---|---|---|
| Freistaat Sachsen | | Tilman Hensch<br>Christian Sanders<br>Nicole Mauche<br>Ulrich Hegerl<br>Markus Loffler |
| LIFE-Leipzig Research Center for Civilization Diseases, University of Leipzig | | Tilman Hensch<br>Christian Sanders<br>Nicole Mauche<br>Ulrich Hegerl<br>Markus Loffler |
| European Union | | Tilman Hensch<br>Christian Sanders<br>Nicole Mauche<br>Ulrich Hegerl<br>Markus Loffler |

The funders had no role in study design, data collection and interpretation, or the decision to submit the work for publication. Open access funding provided by Max Planck Society.

## Author contributions

Alina Studenova, Software, Formal analysis, Investigation, Visualization, Writing – original draft, Writing – review and editing; Carina Forster, Writing – review and editing, Code review; Denis Alexander Engemann, Software, Writing – original draft, Writing – review and editing; Tilman Hensch, Christian Sanders, Nicole Mauche, Ulrich Hegerl, Markus Loffler, Data curation; Arno Villringer, Resources, Writing – original draft, Project administration, Writing – review and editing; Vadim Nikulin, Supervision, Writing – original draft, Project administration, Writing – review and editing

## Author ORCIDs

Alina Studenova ⓘD https://orcid.org/0000-0003-0821-9966
Carina Forster ⓘD https://orcid.org/0000-0001-5827-4387
Denis Alexander Engemann ⓘD https://orcid.org/0000-0002-7223-1014
Tilman Hensch ⓘD https://orcid.org/0000-0003-2696-6017
Christian Sanders ⓘD http://orcid.org/0000-0002-5402-6631
Nicole Mauche ⓘD http://orcid.org/0000-0002-4223-5030
Vadim Nikulin ⓘD https://orcid.org/0000-0001-6082-3859

## Ethics

The LIFE-Adult study was approved by the institutional ethics board of the Medical Faculty of the University of Leipzig (approval numbers 263-2009-14122009, 263/09-ff, 201/17-ek) and was conducted according to the declaration of Helsinki. All participants gave their written informed consent to all measurements and consent to publish.

Reviewer #1 (Public Review): https://doi.org/10.7554/eLife.88367.3.sa1
Reviewer #2 (Public Review): https://doi.org/10.7554/eLife.88367.3.sa2
Author Response https://doi.org/10.7554/eLife.88367.3.sa3

# Additional files

## Supplementary files
• MDAR checklist

## Data availability

The data was collected at the LIFE - Leipzig Research Center for Diseases of Civilization https://www.uniklinikum-leipzig.de/einrichtungen/life/life-forschungszentrum. For data protection reasons, access to the actual study data is only possible via a project agreement https://www.uniklinikum-leipzig.de/einrichtungen/life/life-forschungszentrum/life-datenportal. The application will be

checked and activated by the LIFE Research Center task force. Commercial use of the data is not permitted. The operating instructions are provided to get started with the LIFE data portal https://www.uniklinikum-leipzig.de/einrichtungen/life/Freigegebene%20Dokumente/Projektvereinbarungen/Anleitung%20Datenportal_12062023.pdf. The anonymized peak amplitudes of both P300 and alpha amplitude envelope, standard stimulus ERP and alpha amplitude envelope, and precomputed BSIs are available here https://osf.io/ak5de/. Code for pre-processing and analysis is available at https://github.com/astudenova/p300_alpha (copy archived at *Studenova, 2023*). In addition, two simulated subjects similar to those in the dataset are available to provide the possibility of running the code and obtaining figures. The simulated subjects can be found here https://osf.io/ak5de/.

The following dataset was generated:

| Author(s) | Year | Dataset title | Dataset URL | Database and Identifier |
|---|---|---|---|---|
| Studenova AA, Forster C, Engemann DA, Hensch T, Sander C, Mauche N, Hegerl U, Loeffler M, Villringer A, Nikulin VV | 2023 | Event-related modulation of alpha rhythm explains the auditory P300 evoked response in EEG | https://doi.org/10.17605/OSF.IO/AK5DE | Open Science Framework, 10.17605/OSF.IO/AK5DE |

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

# Appendix 1

**Appendix 1—table 1.** Overview of the previous findings concerning stimulus-related alpha amplitude decrease in an oddball paradigm and similar experiments.
The search for this short review was completed via Google Scholar on 21-09-2021 using keywords: "evoked response", "erp", "erf", "erd", "ers", and on 03-10-2021 using keywords: "P300", "erd", "ers", "eeg", "meg". We picked only research or review papers written in the English language.

| Reference | Experimental paradigm | P300 localisation | P300 latency | Alpha amplitude localization | Alpha amplitude latency |
|---|---|---|---|---|---|
| *Kolev et al., 2001* | passive auditory oddball task | Pz | 300–500ms | Pz | 300–800ms |
| *Yordanova et al., 2001* | active and passive auditory oddball task | Fz, Cz, Pz | average 347ms | Fz, Cz, Pz | average 680ms |
| *Kamarajan et al., 2004* | Go-Nogo task | Pz | 400–800ms | no alpha | - |
| *Kamarajan et al., 2006* | Go-Nogo task | Cz | 300–600ms | Cz | 300–1000ms |
| *Cooper et al., 2008* | passive auditory oddball task | Cz | 280–450ms | all over the cortex | 150–650ms |
| *Digiacomo et al., 2008* | visual stimuli, validly and invalidly cued | Fz, FCz, Cz, Pz | 300–500ms | Fz, FCz, Cz, Pz, P3, O2 | 300–800ms |
| *Ishii et al., 2009* | auditory oddball task | central, non-dipolar pattern (MEG) | 300–400ms | increase in the amplitude over prefrontal cortex (bilateral superior frontal gyrus), decrease in amplitude over sensorimotor cortex (bilateral postcentral gyrus) | 200–600ms |
| *Krämer et al., 2011* | Eriksen flanker task | Fz, Cz, Pz | 300–700ms | C3, C4 | 200–800ms |
| *Peng et al., 2012* | auditory oddball task paired with somatosensory stimuli | Pz | 300–800ms | parietal region | 300–600ms |
| *Barutchu et al., 2013* | audiovisual discrimination task | Oz, O1, O2 | 250–500ms | O1, O2 | 200–800ms |
| *Chen et al., 2013* | hand mental rotation task | Cz, Pz | 300–600ms | CP3 | 300–600ms |
| *Deiber et al., 2013* | visual attention network test | Pz | 300–600ms | P3, P4, Pz | 300–2000ms |
| *Kayser et al., 2014* | auditory novelty oddball task | Pz | 250–700ms | CP2 | 400–800ms |
| *Shou and Ding, 2015* | Stroop-task-switching paradigm | Pz | 300–500ms | Pz | 300–500ms |
| *Zarka et al., 2014* | video stimuli with animation | O1, O2, P3, P4 | 250–500ms | O1, O2, P3, P4 | starts at 300ms |
| *Deiber et al., 2015* | 2-back working memory task | Cz, Pz | 400–1000ms | O2 | 400–800ms |
| *Dong et al., 2015* | Modified Digit Span task | Pz | 300–800ms | parietal region (P3, Pz, P4) | 350–1100ms |
| *Tang et al., 2015* | color-word flanker task | centro-parietal region | 300–600ms | ccipito-parietal region | 300–600ms |
| *Wu et al., 2015* | Stroop-task-switching paradigm | parieto-central region | 300–800ms | fronto-central region | 300–1000ms |
| *Tamura et al., 2016* | auditory presentation of subject's own name and other names | midline | 300–700ms | C4 | 200–600ms |
| *Fabi and Leuthold, 2017* | categorization of pictures with painful and non-painful stimuli | Pz | 400–800ms | sensorimotor cortex | 300–1000ms |
| *Lee et al., 2017* | affective images presentation | posterior region | 300–400ms | posterior region | 100–500ms |

*Appendix 1—table 1 Continued on next page*

*Appendix 1—table 1 Continued*

| Reference | Experimental paradigm | P300 localisation | P300 latency | Alpha amplitude localization | Alpha amplitude latency |
|---|---|---|---|---|---|
| *Leroy et al., 2017* | visual task involving presentation of 2D and 3D images | O2 | 300–600ms | O2 | 100–500ms |
| *López-Caneda et al., 2017* | Go-Nogo task | Fz, Cz, Pz | 300–700ms | Pz | 400–600ms |
| *Vilà-Balló et al., 2017* | auditory novelty oddball task | Pz | 300–800ms | Pz | 300–800ms |
| *Delval et al., 2018* | visual stimuli following gate initiation | Pz | 400–700ms | Cz | no change in the amplitude |
| *Fabi and Leuthold, 2018* | categorization of pictures with painful and non-painful stimuli | Pz | 400–700ms | FC1, FC2, C1, C2, C3, C4, CP1, CP2 | 300–1000ms |
| *Michelini et al., 2018* | four-choice reaction time task | Pz | 300–500ms | parieto-occipital region | 300–1000ms |
| *Liu et al., 2019* | arithmetic problem-solving | O1 | 200–700ms | PO4, PO8 | 200–800ms |
| *Martel et al., 2019* | Go-Nogo task | centro-parietal region | 268–388ms | all channels | 200–600ms |
| *Román-López et al., 2019* | delayed-match to sample task | midline | 300–700ms | midline | 200–1000ms |
| *Faro et al., 2019* | Stroop colour-word test | precuneus | 250–400ms | precuneus | 200–1000ms |
| *Espenhahn et al., 2020* | video viewing while passive tactile stimulation | somatosensory cortex | 270–340 ms | somatosensory cortex | 200–400ms |
| *Kao et al., 2020* | n-back working memory task | Cz, CPz, Pz | 300–800ms | Fz, FCz | 200–900ms |
| *Yu et al., 2020* | hand mental rotation task | Pz | 300–600ms | Pz | 200–600ms |
| *Zhang et al., 2020* | change detection paradigm | ccipital region | 300–600ms | ccipital region | 300–700ms |
| *Nikolin et al., 2021* | n-back working memory task with target stimuli and distractor cues | Fz | 300–500ms | Fz | 300–700ms |
| *Paolicelli et al., 2021* | auditory oddball task | Pz | 280.07–342.67ms | parietal midline | 200–800ms |

