## [Editor Report · eLife assessment]

This is **valuable** study on the mechanistic relationship between two prominent events in post-stimulus EEG: alpha desynchronization and P300 that are known for their slow/relatively late build up. The sample size is substantial. The data are **compelling**, showing that the P300 can be explained by desynchronization of a non-zero mean alpha oscillations over posterior sites through the baseline-shift model, at least partially. This makes a significant contribution to understanding and interpreting P300 generation (and possibly other ERP components) from concurrent changes in brain oscillations, with links to cognition.

---

## [Referee Report · Reviewer #1 (Public Review)]

This EEG study probes the prediction of a mechanistic account of P300 generation through the presence of underlying (alpha) oscillations with a non-zero mean. In this model, the P300 can be explained by a baseline shift mechanism. That is, the non-zero mean alpha oscillations induce asymmetries in the trial-averaged amplitudes of the EEG signal, and the associated baseline shifts can lead to apparent positive (or negative) deflections as alpha becomes desynchronized at around P300 latency. The present paper examines the predictions of this model in a substantial data set (using the typical P300-generating oddball paradigm and careful analyses). The results show that all predictions are fulfilled: the two electrophysiological events (P300, alpha desynchronization) share a common time-course, anatomical sources (from inverse solutions), and covariations with behaviour; plus relate (negatively) in amplitude, while the direction of this relationship is determined by the non-zero-mean deviation of alpha oscillations pre-stimulus (baseline shift index, BSI). This is indictive of a link of the P300 with underlying alpha oscillations through a baseline shift account, and hence that the P300 can be explained, at least in parts, by non-zero mean brain oscillations as they undergo post-stimulus changes.

---

## [Referee Report · Reviewer #2 (Public Review)]

The authors show that event related changes in the alpha band, namely a decrease in alpha power over parieto/occipital areas, explains the P300 during an auditory target detection task. The proposed mechanism by which this happens is a baseline-shift, where ongoing oscillations which have a non-zero mean undergo an event-related modulation in amplitude which then mimics a low frequency event-related potential. In this specific case, it is a negative-mean alpha band oscillation which decreases in power post-stimulus and thus mimics a positivity over parieto-occipital areas, i.e. the P300. The authors lay out 4 criteria that should hold, if indeed alpha modulation generates the P300, which they then go about providing evidence for.

Strengths:

- The authors do go about showing evidence for each prediction rigorously, which is very clearly laid out. In particular I found the 3rd section connecting resting-state alpha BSI to the P300 quite compelling.

- The study is obviously very well-powered.

- Very well-written and clearly laid out. Also the EEG analysis is thorough overall, with sensible analysis choices made.

- I also enjoyed the discussion of the literature.

- The mediation analyses make a convincing argument for behavioural effects being related to BSI also.

Weaknesses:

In general, if one were to be trying to show the potential overlap and confound of alpha-related baseline shift and the P300, as something for future researchers to consider in their experimental design and analysis choices, the four predictions hold well. However, if one were to assert that the P300 is "generated" via alpha baseline shift, even partially, then the predictions either do not hold, or if they do, they are not sufficient to support that hypothesis. Thankfully, the authors no longer make this stronger claim in the revised print. Weaknesses pertaining to the previous draft can be found in the prior review.

In reviewing this paper, I have found the authors have made a convincing case that alpha amplitude modulation potentially confounds with P300 amplitude via baseline shift, and this is a valuable finding.

---

## [Author Response]

The following is the authors’ response to the original reviews.

After thoroughly reviewing the comments and suggestions provided by the reviewers, we have revised our manuscript. We sincerely appreciate the reviewers' constructive approach and valuable feedback. We believe that the edited version of the manuscript is now more comprehensible and reader-friendly. Please find our responses to the comments below.

**Reviewer #1 (Public Review):**
This EEG study probes the prediction of a mechanistic account of P300 generation through the presence of underlying (alpha) oscillations with a non-zero mean. In this model, the P300 can be explained by a baseline shift mechanism. That is, the non-zero mean alpha oscillations induce asymmetries in the trial-averaged amplitudes of the EEG signal, and the associated baseline shifts can lead to apparent positive (or negative) deflections as alpha becomes desynchronized at around P300 latency. The present paper examines the predictions of this model in a substantial data set (using the typical P300-generating oddball paradigm and careful analyses). The results show that all predictions are fulfilled: the two electrophysiological events (P300, alpha desynchronization) share a common time course, anatomical sources (from inverse solutions), and covariations with behaviour; plus relate (negatively) in amplitude, while the direction of this relationship is determined by the non-zero-mean deviation of alpha oscillations pre-stimulus (baseline shift index, BSI). This is indicative of a tight link of the P300 with underlying alpha oscillations through a baseline shift account, at least in older adults, and hence that the P300 can be explained in large parts by non-zero mean brain oscillations as they undergo post-stimulus changes.Specific comments1. The baseline shift model predicts an inverse temporal similarity between alpha envelope changes and P300, confirmed over posterior regions (negative maxima over Pz, Fig 2B). It is therefore intriguing to see in this Figure a very high (positive) correlation in left frontal electrodes. I acknowledge that this is covered in the discussion, but given that this is somewhat unexpected at this point, I suggest providing the readers with a pointer in the Figure legend to this observation and the discussion. Also, I would recommend being more careful with the discussion of this left frontal positive correlation, where a "negative P300" over these areas is mentioned. Given the use of average-referenced sensor data (as opposed to source localized data) and the clear posterior localization of the P300 (Fig 4A), it is likely that what is picked up as "negative ERP potential" over left frontal sites is the posterior P300 forward-projected and inverted through the calculation of the average reference. Accordingly, the interpretation in terms of polarity (positive) of the correlation is likely misleading but what this observation seems to suggest is that other oscillatory processes (than posterior alpha) (e.g. of motor preparation during evidence accumulation) do substantially correlate with the posterior P300 build-up.

We agree that the name P300 should be used rather for positive potential over posterior sites. We edited the text, substituting mentions of “negative P300” for “negative ER”. Also, the following text has been added to the legend of Figure 2:

“Note the positive correlation between the low-frequency signal and the alpha amplitude envelope over central sites. Due to the negative polarity of ER over the fronto-central sites, such correlation may still indicate a temporal relationship between the P300 process and oscillatory amplitude envelope dynamics (due to the use of a common average reference). However, it cannot be entirely excluded that additional lateralized response-related activity contributes to this positive correlation (Salisbury et al., 2001).”

1. Parts of the conclusions are based on a relationship between alpha-amplitude modulation and size of P300-amplitude (amplitude-amplitude) using data binning (illustrated in Fig 3) and the bins seem to include different participants, rather than trials. As this is an analysis of EEG data, I wonder how much of this relationship can be explained by a confound of skull thickness (or other individual differences in anatomy picked up with the scalp measures such as gyral folding patterns and current source orientations etc). E.g. those with thicker/thinner skulls are expected to show less/more of a modulation in all signals. This could be ruled out by relating the bins in alpha modulation not to the P300 but to another event that does not coincide in time with the alpha changes (e.g. P100), where no changes across bins would be expected.

We are grateful for the suggestions on confound estimation. We repeated the analysis of binning of alpha rhythm amplitude normalised change in relation to early ER, which in our auditory paradigm was N100. The largest change in the alpha amplitude occurs later in the poststimulus window, but that does not necessarily mean that the activity in the window right after the stimulus onset is unaffected. As can be seen in Figure 3 (t-statistics between alpha bins), there is already a significant difference around 100 ms over the central regions of the scalp. For this plot, the broadband data was filtered from 0.1 to 3 Hz, thus assessing only changes in low-frequency signals. We repeated the same analysis for broadband data (0.1–45 Hz) and also observed a significant difference between two extreme bins around 100 ms over the central region (Figure S5A). However, if we filter the signal from 4 to 45 Hz, these significant differences almost completely disappear (only electrode TP9 was significant; Figure S5B). Importantly, this range (4–45 Hz) includes the frequency of N100, which is typically in the alpha range. It means that the differences in N100 are riding on top of the baseline shift created by an unfolding alpha amplitude decrease. When this low-frequency baseline shift was removed, significant differences were no longer visible. This is an indication that differences in P300 amplitude between alpha bins are restricted to the low-frequency range and are not propagated to other ERs with higher frequency content.

We added Figure S5 to the Supplementary material and introduced it in the main text, the Results section, as follows:

“The cluster within the earlier window (100–200 ms) over central regions (Figure 3C) possibly reflects the previously shown effect of prestimulus alpha amplitude on earlier ERs (Brandt et al., 1991, Babiloni et al., 2008) but may also be a manifestation of BSM. We tested this assumption for early ER, which in our auditory task was N100. We repeated the binning analysis for broadband data (0.1–45 Hz) and also observed a significant difference between two extreme bins around 100 ms over the central region (Figure S5A). However, if we filter the signal from 4 to 45 Hz (the range that includes the frequency of N100 but not low-frequency baseline shifts), these significant differences almost completely disappear (only electrode TP9 was significant; Figure S5B). It means that the difference in N100 amplitudes over frontal sites is driven by the baseline shift created by an unfolding alpha amplitude decrease. The significant difference at the TP9 electrode possibly reflects a genuine physiological effect of alpha rhythm amplitude on the excitability of a neuronal network and, as a consequence, on the amplitude of ER (as opposed to the baseline-shift mechanism, where the alpha rhythm doesn’t affect the amplitude of ER but creates an additional component of ER; Iemi et al. 2019).”

1. Related to the above: I assume it can be ruled out that the relationship between baseline-shift index and P300 amplitude (also determined through binning, Fig 6) could be influenced by the above-mentioned confounds, given the inverse relationship?

As in previous studies alpha rhythm power was found to depend on the size of the head (Candelaria-Cook et al., Cerebral Cortex, 2022), we agree that the contribution of this confounding factor should be estimated (and we did estimate it). However, we would like to point out that we looked into dependencies based on ratios, which eliminates absolute units potentially being affected by head size, skull thickness, etc. For instance, the baseline-shift index is estimated as the Pearson correlation coefficient between the alpha rhythm envelope and low-frequency signal during the resting state. Therefore, multiplying the alpha amplitude envelope by an arbitrary scale would not cause the correlation to change. Nonetheless, for a subset of participants (1034 participants, mean age 69.8 years, 496 female), we had MRI data, from which we extracted total intracranial volume. For each electrode, we computed the Pearson correlation between the variable of interest and total intracranial volume. Variables of interest were the peak amplitude of P300, the attenuation-peak amplitude of alpha rhythm, alpha rhythm normalised amplitude (computed as Apost −Apre Apre ∗100%), and the magnitude of the baseline shift index (BSI). The p-value was set at Bonferroni corrected 0.05. For P300, only one electrode, namely C4, demonstrated a significant correlation of –0.10. However,the C4 electrode is outside of the typical electrode range for P300. For alpha envelope amplitude, significant correlations were observed all over the head (19 out of 31 electrodes, maximum at Cz), and a larger total intracranial volume was related to a higher amplitude of alpha rhythm.

Candelaria-Cook et al. (Cerebral Cortex, 2022) showed a similar association in longitudinal data from children and adolescents, but the increase in alpha rhythm power in that study might have been due to additional factors beyond a growing head. Conversely, normalised alpha amplitude showed no significant correlations. Similarly, the absolute value of BSI did not correlate significantly with total intracranial volume at any electrode. Overall, only alpha amplitude shows a prominent correlation to total brain volume, thus reducing the concern that head size may be a confound.

1. This study is based on a sample of older participants. One wonders to what extent this is needed to reveal the alpha-P300 relationships (e.g. more variability in this population than in younger controls), and/or whether other mechanisms may be at play across the lifespan.

Our study is indeed based on a sample of older participants. However, in our previous study (Studenova et al., PLOS Comp Bio, 2022), we compared young and elderly participants using resting-state data. There, we measured the baseline-shift index (BSI) at rest, and BSI serves as a proxy for baseline shifts present in the task-based data (under the assumptions of the baseline-shift mechanism, ER is in essence a baseline shift). We found that BSIs for elderly participants were smaller in comparison to those for young participants. Yet, the distribution of BSI values across the scalp (as in Figure 6A) was similar between the two age groups.

Additionally, we observed that larger alpha rhythm power was positively correlated with the magnitude of BSI, but only for younger participants, which points out possible difficulties arising from the fact that elderly people have reduced alpha power. Therefore, we believe that for a sample of young participants, the results should not be different.

1. Legend to Figure 6: sentence under A: "A positive deflection of P300 at posterior sites coincides with a decrease in alpha amplitude, a case that corresponds to negative mean oscillations." I find this sentence at this place in the legend confusing, as Fig 6A seems to illustrate the BSI only (not yet any relationship?).

We expanded the text in the legend with this paragraph:

“BSI serves as a proxy for the relation between ER polarity and the direction of alpha amplitude change (Nikulin et al., 2010). Here, we observe predominantly negative BSIs (and thus negative mean oscillations) at posterior sites, which indicates the inverted relation between P300 and alpha amplitude change. Indeed, in the task data, a positive deflection of P300 at posterior sites coincides with a decrease in alpha amplitude.”

1. Page 4: repetition of "has been" "has been" one after each other in the text We are thankful for this catch. We removed the repetition.
**Reviewer #2 (Public Review):**
The authors attempt to show that event-related changes in the alpha band, namely a decrease in alpha power over parieto/occipital areas, explain the P300 during an auditory target detection task. The proposed mechanism by which this happens is a baseline-shift, where ongoing oscillations which have a non-zero mean undergo an event-related modulation in amplitude which then mimics a low frequency event-related potential. In this specific case, it is a negative-mean alpha-band oscillation that decreases in power post-stimulus and thus mimics a positivity over parieto-occipital areas, i.e. the P300. The authors lay out 4 criteria that should hold if indeed alpha modulation generates the P300, which they then go about providing evidence for.Strengths:The authors do go about showing evidence for each prediction rigorously, which is very clearly laid out. In particular, I found the 3rd section connecting resting-state alpha BSI to the P300 quite compelling.The study is obviously very well-powered.Very well-written and clearly laid out. Also, the EEG analysis is thorough overall, with sensible analysis choices made.I also enjoyed the discussion of the literature, albeit with certain strands of P300 research missing.Weaknesses:In general, if one were to be trying to show the potential overlap and confound of alpha-related baseline shift and the P300, as something for future researchers to consider in their experimental design and analysis choices, the four predictions hold well enough. However, if one were to assert that the P300 is "generated" via alpha baseline shift, even partially, then the predictions either do not hold, or if they do, they are not sufficient to support that hypothesis. This general issue is to be found throughout the review. I will briefly go through each of the predictions in turn:1. The matching temporal course of alpha and P300 is not as clear as it could be. Really, for such a strong statement as the P300 being generated by alpha modulation, one would need to show a very tight link between the signals temporally. There are many neural and ocular signals which occur over the course of target detection paradigms: P300, alpha decrease, motor-related beta decrease, the LRP, the CNV, microsaccade rate suppression etc. To specifically go above and beyond this general set of signals and show a tighter link between alpha and P300 requires a deeper comparison. To start, it would be a good idea to show the signals overlapping on the same plot to really get an idea of temporal similarity. Also, with the P300-alpha correlation, how much of this correlation is down to EEG-related issues such as skull thickness, cortical folding, or cognitive issues such as task engagement? One could perhaps find another slow wave ERP,e.g. the Lateralised Readiness Potential, and see if there is a similar strength correlation. If there is not, that would make the P300 relationship stand out.

Thank you for this comment. In our study, we outline the prerequisites for the baseline-shift mechanism (BSM) and show how they hold for the obtained data. Overall, for all the prerequisites, the evidence could be found in favour of BSM. However, as it is the case for all EEG/MEG data, the non-invasive nature of the data puts constraints on the interpretation of the results. In order to specifically address the points raised by the reviewer about the results, we provide additional information about the overlap (Figure 2) and non-specific anatomical parameters.

The baseline-shift mechanism makes a general prediction about the generation of some ERs(those that coincide with a change in oscillatory amplitudes). The fact that neuronal oscillations (especially alpha oscillations) are modulated in almost any task indicates that other ERs can also contain a contribution from the baseline-shift mechanism. In our study, it is plausible that several sources of alpha oscillations orchestrated several ER components that appeared on the scalp after the presentation of a target stimulus. Due to the substantial spatial mixing and temporal overlap, it is difficult to disentangle the processes indexing perceptual, memory, or motor functions. However, currently, we are working on showing that the readiness potential (movement related potential) in the classical Libet’s paradigm also complies with the baseline-shift mechanism.

Concerns about confounds such as skull thickness are valid; therefore, we performed additional analysis. For a subset of participants (1034 participants, mean age 69.8 years, 496 female), we had MRI data, from which we extracted total intracranial volume. We tested the correlation between total intracranial volume and several variables of interest: the peak amplitude of P300, the attenuation-peak amplitude of alpha rhythm, alpha rhythm normalised change, and the magnitude of the baseline shift index (BSI). For P300 amplitude, only the C4 electrode showed a significant correlation of –0.10. For alpha envelope amplitude, there were significant correlations all over the head (19 out of 31 electrodes, maximum at Cz). The correlations showed that a larger total intracranial volume was related to a higher amplitude of alpha rhythm. For a normalised change in alpha amplitude, we observed no significant correlations. Similarly, the absolute value of BSI did not correlate significantly with total intracranial volume at any electrode. Overall, alpha amplitude indeed shows a prominent correlation to total brain volume, but none of the relational variables (normalised amplitude change, BSI) show any correlation.

In Figure 3, it is clear that alpha binning does not account for even 50% of the variance of P300 amplitude. Again, if there is such a tight link between the two signals, one would expect the majority of P300 variance to be accounted for by alpha binning. As an aside, the alpha binning clearly creates the discrepancy in the baseline period, with all alpha hitting an amplitude baseline at approx. 500ms. I wonder if could you NOT, in fact, baseline your slow wave ERP signal, instead using an appropriate high pass filter (see "EEG is better left alone", Arnaud Delorme, 2023) and show that the alpha binning creates the difference in ERP at the baseline which then is reinterpreted as a P300 peak difference after baselining.

The difference in the baseline window for alpha rhythm amplitude is indeed prominent (Figure R1A,B), so we proceed with the suggested analysis. Before anything else, we would like to reiterate that the baseline correction per se does not generate ER; it just moves the whole curve (in the pre- and poststimulus intervals) up and down. Firstly, we repeated the analysis without baseline correction (filter 0.1–3 Hz) and still observed the difference in P300 amplitude across bins (Figure R1D). Moreover, based on cluster-based permutation testing, ERs in the two most extreme bins were not significantly different in the prestimulus window. However, when we opt for no baseline correction, there will still be a baseline, namely, the average of the signal will be zero within a filtering window (e.g., 10 sec for a high-pass filter at 0.1 Hz). Thus, secondly, we computed an ER but with the baseline in the poststimulus window (400–600 ms; Figure R1E). In this case, the difference between bin 1 and bin 5 (for the prestimulus interval) in the window before 0 ms was significant in the posterior regions. The differences in the baseline are perceived as being smaller than the differences in alpha amplitude. This can be attributed to the fact that there are other low-frequency processes in the EEG signal that are different from alpha baseline shifts. Additionally, P300 in bin 1 in comparison with P300 in bin 5 is significantly different in shape (Figure R1C). This can be an indication of overlapping components; namely, for bin 5 (where alpha amplitude change is the highest), associated baseline shift dominates, and for bin 1 (where alpha amplitude change is the smallest), associated baseline shift is hidden behind other components. We believe that this proposed analysis demonstrates the intuition behind the baseline-shift mechanism: the baseline shift is generated due to a change in the oscillatory amplitude; and the change is simply the difference between two time points.

**Author response image 1. sa3fig1:** The difference in the strength of alpha amplitude modulation correlates with the difference in P300 amplitude. A. The alpha rhythm amplitude was binned according to the percentage of change. The bins were the following: (66, –25), (–25, –37), (–37, –47), (–47, –58), (–58,–89) % change. A is identical to Figure 3A, main text. B. The alpha rhythm amplitude is multiplied by –1 and evened within the prestimulus window. This may be an approximation for baseline shifts in the low-frequency signal. C. P300 responses are sorted into the corresponding bins. The C is identical to Figure 3B, main text. D. P300 are obtained without applying a baseline correction and are sorted into the corresponding bins. The difference in peak amplitude of P300 remains visible and significant. E. P300 is baselined at 400–600 ms. As a consequence, there are significant differences in the prestimulus window.

1. The topographies are somewhat similar in Figure 4, but not overwhelmingly so. There is a parieto-occipital focus in both, but to support the main thesis, I feel one would want to show an exact focus on the same electrode. Showing a general overlap in spatial distribution is not enough for the main thesis of the paper, referring to the point I make in the first paragraph re Weaknesses. Obviously, the low density montage here is a limitation. Nevertheless, one could use a CSD transform to get more focused topographies (see https://psychophysiology.cpmc.columbia.edu/software/csdtoolbox/), which apparently does still work for lower-density electrode setups (see Kayser and Tenke, 2006).

As we mentioned in our provisional response, we believe that we would not benefit from using CSD. First, the CSD transform is a spatial high-pass filter, and, hence, it is commonly used for spatially localised activities. In our case, we have two activities—P300 and alpha amplitude decrease—that are widespread with low spatial frequency, and we believe that applying CSD is not helpful. Second, CSD is more sensitive to surface sources that emanate from the crowns of gyri. For activity in the P300 window, there is a possibility that sources are localised within the longitudinal fissure. Third, as we completely agree that low density montage is a limitation, we used source reconstruction with eLoreta (Figure 5) to clarify the spatial localisation of the potential source of P300 and alpha amplitude change, which indeed shows a considerable spatial overlap.

1. Very nice analysis in Figure 6, probably the most convincing result comparing BSI in steady state to P300, thus at least eliminating task-related confounds.1. Also a good analysis here, wherein there seem to be similar correlation profiles across P300 and alpha modulation. One analysis that would really nail this down would be a mediation analysis (Baron and Kenny, 1986; https://davidakenny.net/cm/mediate.htm), where one could investigate if e.g. the relationship between P300 amplitude and CERAD score is either entirely or partially mediated by alpha amplitude. One could do this for each of the relationships. To show complete mediation of P300 relationship with a cog task via alpha would be quite strong.

We agree that mediation analysis better suits the purpose of our claim. We added this analysis to the edited version of the manuscript. Additionally, we became concerned that the total alpha power effect may be driving the correlation. Therefore, we used alpha amplitude change in percentage instead of the absolute values of the amplitude. Significant mediation was present only for attention and executive scores.

In the updated version of the manuscript, the Methods section reads as follows:

“The correlation between cognitive scores (see Methods/Cognitive tests) and the amplitude and latency of P300 and alpha oscillations was calculated with linear regression using age as a covariate (R lme4, Bates et al., 2015). To estimate what proportion of the correlation between P300 and cognitive score is mediated by alpha oscillations, we used mediation analysis (Baron et al., 1986; R mediation, Tingley et al, 2014). First, we estimated the effect of P300 on the cognitive variable of interest (total effect, cogscore ~ P300+age). Second, we computed the association between P300 and alpha oscillations (the effect on the mediator, alpha ~ P300).Third, we run the full model (the effect of the mediator on the variable of interest, cogscore ~P300+alpha+age). Lastly, we estimated the proportion mediated.”

The Results section reads as follows:

“Stimulus-based changes in brain signals are thought to reflect cognitive processes that are involved in the task. A simultaneous and congruent correlation of P300 and alpha rhythm to a particular cognitive score would be another evidence in favour of the relation between P300 and alpha oscillations. Moreover, if thus found, the correlation directions should correspond to the predictions according to BSM. Along with the EEG data, in the LIFE data set, a variety of cognitive tests were collected, including the Trail-making Test (TMT) A&B, Stroop test, and CERADplus neuropsychological test battery (Loeffler et al., 2015). From the cognitive tests, we extracted composite scores for attention, memory, and executive function (Liem et al., 2017, see Methods/Cognitive tests) and tested the correlation between composite cognitive scores vs. P300 and vs. alpha amplitude modulation. The scores were available for a subset of 1549 participants (out of 2230), age range 60.03–80.01 years old. Cognitive scores correlated significantly with age (age and attention: −0.25, age and memory: −0.20, age and executive function: −0.23). Therefore, correlations between cognitive scores and electrophysiological variables were evaluated, regressing out the effect of age. To rule out the possibility of a absolute alpha power association with cognitive scores, for this analysis, we used alpha amplitude normalised changecomputed as Apost −Apre Apre ∗100%, where Apost is at the latency of strongest amplitudedecsease. Computed this way, negative alpha amplitude change would correspond to a more pronounced decrease, i.e., stronger oscillatory response.

To increase the signal-to-noise ratio of both P300 and alpha rhythm, we performed spatial filtering (see Methods/Spatial filtering, Figures 7B,C). Following this procedure, both P300 and alpha latency, but not amplitude, significantly correlated with attention scores (Figure 7A, left column). Larger latencies were related to lower attentional scores, which corresponded to a longer time-to-complete of TMT and Stroop tests and hence poorer performance. The proportion of correlation between P300 latency and attention, mediated by alpha attenuation peak latency, is 0.12. Memory scores were positively related to P300 amplitude and negatively to P300 latency (Figure 7A, middle column). The direction of correlation is such that higher memory scores, which reflected more recalled items, corresponded to a higher P300 amplitude and an earlier P300 peak. The association between alpha rhythm parameters and memory scores is not significant, but it goes in the same direction as the association for P300. Executive function (Figure 7A, right column) were related significantly to both P300 and alpha amplitude latencies. The proportion of correlation between P300 latency and attention, mediated by alpha attenuation peak latency, is 0.14. Overall, the direction of correlation is similar for P300 and alpha oscillations, as expected for BSM. Moreover, the direction of correlation is consistent across cognitive functions.

And an additional paragraph in the Discussion:

“The mediation analysis showed that the modulation of alpha oscillations only partially explained the correlation between P300 and cognitive variables. This, in general, corresponds to the idea that not the whole P300 but only its fraction can be explained by the changes in the alpha amplitudes. Figure 5 shows that alpha oscillations change not only in the cortical areas where P300 is generated; therefore, we cannot expect a complete correspondence between the two processes. Moreover, since cognitive tests and EEG recordings were performed at different time points, the associations between the cognitive variables and EEG markers are expected to be rather weak and to reflect only some neuronal processes common to P300, alpha rhythm, and tasks. For these reasons, a complete mediation of one EEG variable through another EEG variable in the context of a separate cognitive assessment cannot be expected.”

One last point, from the methods it appears that the task was done with eyes closed? That is an extremely important point when considering the potential impact of alpha amplitude modulation on any other EEG component due to the well-known substantial increase in alpha amplitude with eyes closed versus open. I wonder, would we see any of these effects with eyes opened?

The task was auditory and was indeed conducted in an eyes-closed state. In an eyes-closed state, alpha rhythm amplitude in the occipital regions shows a prominent increase. However, we believe that in our case, it was neither an advantage nor a disadvantage. First, occipital sources of alpha rhythm that demonstrate an increase in amplitude are not likely to be those sources that attenuate as a reaction to a target tone. The source reconstruction of alpha rhythm amplitude change (although with a limited number of channels) displayed widespread regions with a prominent decrease on the posterior midline, including the precuneus and posterior cingulate cortex (which contain polymodal association areas; Leech et al., Brain, 2014; Al-Ramadhani et al., Epileptic Disord, 2021). Second, in our previous study, we tested resting-state data with both eyes-closed and eyes-open conditions. There, we computed the baseline-shift index (BSI), which serves as an approximation for estimating if oscillations have a non-zero mean. We found no significant difference between the eyes-open and eyes-closed states in terms of the absolute value of the BSI. Moreover, the average distribution of BSIs on the scalp was the same for both conditions.

Overall, there is a mix here of strengths of claims throughout the paper. For example, the first paragraph of the discussion starts out with "In the current study, we provided comprehensive evidence for the hypothesis that the baseline-shift mechanism (BSM) is accountable for the generation of P300 via the modulation of alpha oscillations." and ends with "Therefore, P300, at least to a certain extent, is generated as a consequence of stimulus-triggered modulation of alpha oscillations with a non-zero mean." In the limitations section, it says the current study speaks for a partial rather than exhausting explanation of the P300's origin. I would agree with the first part of that statement, that it is only partial. I do not agree, however, that it speaks to the ORIGIN of the P300, unless by origin one simply means the set of signals that go to make up the ERP component at the scalp-level (as opposed to neural origin).

We have edited parts of the manuscript that have overly exuberant claims. However, we would argue further that alpha rhythm amplitude change does partially explain P300 origin. When a stimulus is being processed by the neuronal network, some part of this network presumably breaks from synchronous oscillation mode. Hence, on the scalp, we observe a decrease in oscillatory amplitude. According to the baseline-shift mechanism (BSM), this stimulus-related decrease in the amplitude generates the baseline shift in the frequency range of modulation (under 3 Hz for alpha rhythm). The P300 component that is explained by alpha rhythm amplitude modulation is, in essence, a baseline shift. Therefore, the origin of a part of P300 is the oscillating network that was pushed out of its synchronous oscillating regime.

Again, I can only make these hopefully helpful criticisms and suggestions because the paper is very clearly written and well analysed. Also, the fact that alpha amplitude modulation potentially confounds with P300 amplitude via baseline shift is a valuable finding.Specific comments:Perhaps give a brief overview of the task involved at the start. I know it is not particularly relevant, but I think necessary for those unfamiliar with cog tasks.

We added a short description of a task in the Introduction section.

“In this data set, the experimental task was an auditory oddball paradigm. Participants would hear tones, one type of which—the target tone—would occur in only 12% of trials. Target tones elicit both P300 and the modulation of the alpha amplitude. ”